# Sleep fragmentation exacerbates myocardial ischemia–reperfusion injury by promoting copper overload in cardiomyocytes

Na Chen [1], Lizhe Guo [1], Lu Wang[1], Sisi Dai[1], Xiaocheng Zhu[1] & E. Wang [1,2] ✉

Sleep disorders increase the risk and mortality of heart disease, but the brain-heart interaction has not yet been fully elucidated. Cuproptosis is a copper-dependent type of cell death activated by the excessive accumulation of intracellular copper. Here, we showed that 16 weeks of sleep fragmentation (SF) resulted in elevated copper levels in the male mouse heart and exacerbated myocardial ischemia–reperfusion injury with increased myocardial cuproptosis and apoptosis. Mechanistically, we found that SF promotes sympathetic overactivity, increases the germination of myocardial sympathetic nerve terminals, and increases the level of norepinephrine in cardiac tissue, thereby inhibits VPS35 expression and leads to impaired ATP7A related copper transport and copper overload in cardiomyocytes. Copper overload further leads to exacerbated cuproptosis and apoptosis, and these effects can be rescued by excision of the sympathetic nerve or administration of copper chelating agent. Our study elucidates one of the molecular mechanisms by which sleep disorders aggravate myocardial injury and suggests possible targets for intervention.

Sleep disorders have been reported to be associated with increased morbidity and mortality of coronary heart disease[1,2]. A systematic review including 15 studies (24 cohort samples) with 474,684 participants pointed out that people with a short duration of sleep have a greater risk of coronary heart disease (RR 1.48, 95% CI 1.22–1.80, $p < 0.0001$)[2]. Data from experimental animal models also confirmed the detrimental effect of sleep deprivation on cardiac injury[3–5]. However, the underlying molecular mechanisms responsible for this effect have not been fully elucidated and require further investigation.

Myocardial injury is related to ion homeostasis of cardiomyocytes. Copper homeostasis is crucial for maintaining various physiological functions in organisms[6,7], such as mitochondrial energy production[8], redox homeostasis[9], tyrosine and neurotransmitter metabolism[10,11], and extracellular matrix remodeling[12]. Many studies have shown that copper deficiency can lead to increased collagen deposition and increased myocardial fibrosis[13], cause impaired antioxidant defense and increased susceptibility to oxidative stress[14], and

affect angiogenesis[15,16]. Moreover, evidence has proven the detrimental effect of copper overload. Cellular copper overload activates the apoptosis pathway via endogenous and exogenous pathways[17–19]. On the other hand, copper overload also leads to cuproptosis, a recently newly identified, copper-dependent type of cell death characterized by loss of iron-sulfur cluster protein, reduced levels of lipoylated proteins and increased oligomerization of DLAT[20]. However, few studies have investigated the role of copper overload or cuproptosis in cardiovascular disease.

Sympathetic hyperactivity is an important pathological feature of both sleep disorders[21] and cardiovascular diseases[22,23]. However, the correlation between sympathetic signal regulation and cardiac copper homeostasis remains to be confirmed. We established a mouse model of myocardial ischemia–reperfusion injury (MI/RI, see Table 1 for all acronyms in the manuscript) aggravated by sleep fragmentation (SF) and a copper overload model of HL-1 cells induced by norepinephrine (NE) to elucidate the effects of SF on

[1]Department of Anesthesiology, Xiangya Hospital, Central South University, Changsha, China. [2]National Clinical Research Center for Geriatric Disorders (Xiangya Hospital), Changsha, China. ✉e-mail: ewang324@csu.edu.cn

**Table 1 | Non-standard abbreviations and acronyms**

| Acronyms | Nonstandard abbreviations |
|---|---|
| MI/RI | Myocardial ischemia-reperfusion injury |
| SF | Sleep fragmentation |
| TTC | 2,3,5-triphenyl tetrazolium chloride |
| TH | Tyrosine hydroxylase |
| SCGx | Superior cervical ganglionectomy |
| TTM | Tetrathiomolybdate |
| CS 1 | Coppersensor-1 |
| GFP | Green fluorescent protein |
| TUNEL | TdT-mediated dUTP nick-end labeling |
| Lip-DLAT | Lipoylated dihydrolipoamide S-acetyltransferase |
| Lip-DLST | Lipoylated dihydrolipoamide S-succinyltransferase |
| WGA | Wheat germ agglutinin |
| BCL-2 | B-cell lymphoma-2 |
| NE | Norepinephrine |
| EPI | Epinephrine |
| CORT | Cortisol |
| VPS35 OE | Vps35 overexpression |
| PLZ | Phenelzine |
| MAOI | Monoamine oxidases inhibitor |
| GSH | Reduced glutathione |
| GSSG | Oxidized glutathione |

cardiac copper metabolism and the underlying molecular mechanism and to confirm the therapeutic effect of interventions targeting cardiac copper overload in a model of cardiac ischemia–reperfusion injury aggravated by SF. Our study showed that the regulation of cardiac copper homeostasis by sympathetic nerve activity is a mechanism of brain-heart interactions.

## Results
### Chronic SF exacerbated MI/RI and induced cardiomyocyte cuproptosis and apoptosis
To investigate how sleep disorders deteriorate MI/RI, we subjected C57BL/6 mice to chronic SF for 16 weeks before we established the MI/RI model (Fig. 1A). We found that on the third day after MI/RI, mice with SF showed a significant decrease in cardiac function compared with that of control mice (Fig. 1B–F, Supplementary Fig. S1A–D). Further investigation revealed that after MI/RI, mice with SF had longer left ventricular weight/tibia lengths, higher lung weight/body weight ratios (Supplementary Fig. S1E, F), larger infarct areas (Fig. 1G, H), and more TUNEL$^+$ cells in the infarcted and marginal area of myocardium (Fig. 1I, J). We then sought to identify the cell death pathway that was altered in our experimental model and found that SF significantly enhanced cuproptosis (Fig. 2A–I, Supplementary Fig. S2A–D) and apoptosis after MI/RI (Supplementary Fig. S2E, J–O) but not necroptosis (Supplementary Fig. S2F–H) or ferroptosis (Supplementary Fig. S2I, J, P). Specifically, after MI/RI, control mice had restricted signals in only the ischemic zone in the lower segment of the heart, whereas mice with SF had significantly increased DLAT aggregation in the MI/RI region, which indicated the activation of cuproptosis (Fig. 2A, B). The activation of cuproptosis was also confirmed by elevated protein levels of DLAT oligomers and HSP70 in the myocardium of mice with SF + IR (Fig. 2C, D, I). Mice with SF + IR exhibited reduced expression of iron-sulfur cluster proteins, such as FDX1, SDHB, ACO2 and LIAS (Fig. 2C, H, Supplementary Fig. S2A–D), as well as lipoylated proteins, such as lip-DLAT and Lip-DLST (Fig. 2C, F, G), which demonstrates that severe copper overload might occur in the myocardium of mice with SF + IR.

### Chronic SF induced myocardial copper overload
After investigating the metal ion content in the myocardium, it was found that copper levels significantly increased in the myocardium of mice with SF before and after MI/RI (Fig. 3A), while there were no significant differences in the levels of other essential metals (Supplementary Fig. S3A, B). As reported in previous studies[24–26], Slc31a and ATP7A are mainly responsible for the transport of copper ions into and out of cells. A study by our group revealed that downregulation of the copper transporter Slc31a indicated chronic copper overload in the myocardium of mice with SF (Fig. 3C)[27]. We found that the upregulation of ATP7A[28] might also have a compensatory effect on chronic copper overload (Fig. 3D–F). Immunofluorescence staining showed distinct spatial distribution patterns of ATP7A in the myocardium between control mice and mice with SF, which also supports the existence of copper overload in the myocardium in mice with SF. In the control group, ATP7A accumulated around the nucleus, while in mice with SF, ATP7A was free in the cytoplasm (Fig. 3G). Therefore, we confirmed the existence of copper overload in the myocardium of mice with SF, but the underlying mechanism remained to be elucidated.

### Sympathetic hyperactivity mediated myocardial copper overload in mice with SF
Sympathetic hyperactivity is a common pathological feature of sleep disorders[21], and it often affects cardiovascular function. By performing cardiac electrophysiological tests, we found that SF significantly increased the resting heart rate (HR) and decreased the high frequency power spectral density (HF), indicating cardiac sympathetic hyperactivity and a reduction in parasympathetic innervation in mice with SF (Fig. 3H–J, Supplementary Fig. S3C). In addition, immunofluorescence staining revealed a significant increase in the abundance of sympathetic nerve terminals in mice with SF (Fig. 3K–M). Furthermore, compared to mice in the control group, mice in the SF group had significantly increased levels of myocardial NE, the primary neurotransmitter released at sympathetic nerve terminals, and plasma epinephrine (EPI) (Fig. 3N, Supplementary Fig. S3D), and there were no differences in myocardial EPI, plasma NE or corticosterone levels between the two groups (Fig. 3O, Supplementary Fig. S3E, F). Therefore, due to sympathetic hyperactivity, a high level of myocardial NE might induce copper overload in the myocardium.

As in previous studies, we utilized coppersensor-1 (CS1) as a selective inducible fluorescence sensor for imaging intracellular copper[29], and we established an experimental model in vitro by treating HL-1 cells with $CuCl_2$ to investigate whether and how NE could affect copper transport (Fig. 4A). The immunofluorescence results indicated that $CuCl_2$ treatment could activate intracellular copper transport and mobilize intracellular ATP7A for copper vesicular transport (Fig. 4B) and that NE treatment could significantly increase intracellular copper levels and induce block copper transport, with more ATP7A colocalized with the CS1 signal (Fig. 4B, C), and significantly activate intracellular cuproptosis (Fig. 4D–K). Since high concentrations of catecholamines can induce increased reactive oxygen species formation and aldehyde concentration through monoamine oxidases (MAOs), and SF mice were experiencing increased oxidative stress (Supplementary Fig. S4A–C). We treated HL-1 cells with NE in the presence of an MAO inhibitor and found that the increased cellular copper overload and cuproptosis brought about by NE was not attenuated by the MAO inhibitor (Supplementary Fig. S4D–O). These results showed that NE treatment significantly activated intracellular copper overload and cuproptosis, which was independent of MAOs, indicating an important role of sympathetic signaling in myocardial copper overload.

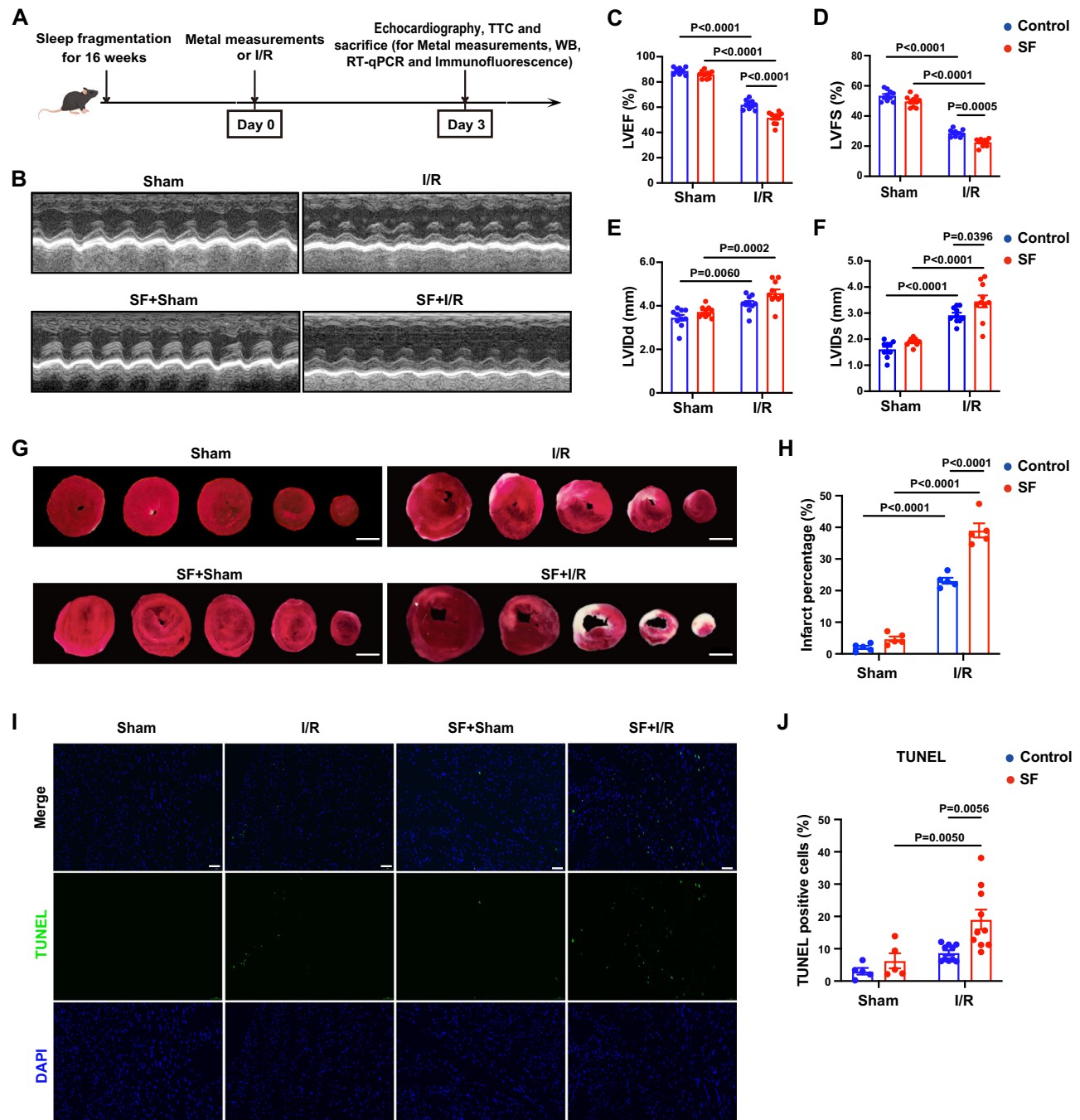

**Fig. 1 | Chronic sleep fragmentation (SF) exacerbates myocardial ischemia–reperfusion injury (MI/RI) in mice. A** Schematic of the experimental protocol used to establish chronic SF-exacerbated MI/RI. I/R, ischemia–reperfusion. Cartoon mouse images was drawn by Figdraw (https://www.figdraw.com/#/). **B–F** Mice with SF had worse cardiac function after MI/RI. **B** Representative M-mode echocardiographic changes. **C–F** Left ventricular ejection fraction (LVEF). Left ventricular fractional shortening (LVFS). Left ventricular internal diameter, systolic (LVIDs) and left ventricular internal diameter, diastolic (LVIDd) were assessed by echocardiography in mice with SF and control mice before and after MI/RI (*n* = 10 mice per group). **G, H** Mice with SF had larger infarct sizes. **G** Representative images of TTC staining of the myocardium. The white area indicates the necrotic tissue in the infarct area, and the red area indicates the non-infarcted area. Scale bar, 2 mm. **H** The percentage of infarct area was determined by TTC staining (*n* = 5 mice per group). **I, J** Mice with SF had increased rates of cardiomyocyte death after MI/RI. **I** Representative images of TdT-mediated dUTP nick-end labeling (TUNEL) staining of cardiomyocytes in the infarcted and marginal area by confocal microscopy. Scale bar, 50 μm (*n* = 5,10 mice for sham and I/R group). **J** Averaged data on TUNEL-positive cells were calculated as the ratio of TUNEL-positive nuclei to DAPI-stained nuclei (*n* = 5, 10 mice for sham and I/R group). Data are presented as the mean ±s.e.m. Statistical analysis was performed using one-way ANOVA.

## VPS35-related impairment of copper ion transport mediates copper overload via sympathetic hyperactivity

To further explore the mechanism underlying myocardial copper overload, we examined the expression of genes that have been reported to be members of transport networks for intracellular copper ions[25,30–32] (Fig. S5A–S5F) and found that the expression of VPS35 was inhibited in the myocardium of both mice with SF and HL-1 cells treated with NE (Fig. 5A–C). We hypothesized that inhibition of VPS35 expression was the key cause of the impaired intracellular copper transport induced by NE treatment. Therefore, we constructed HL-1

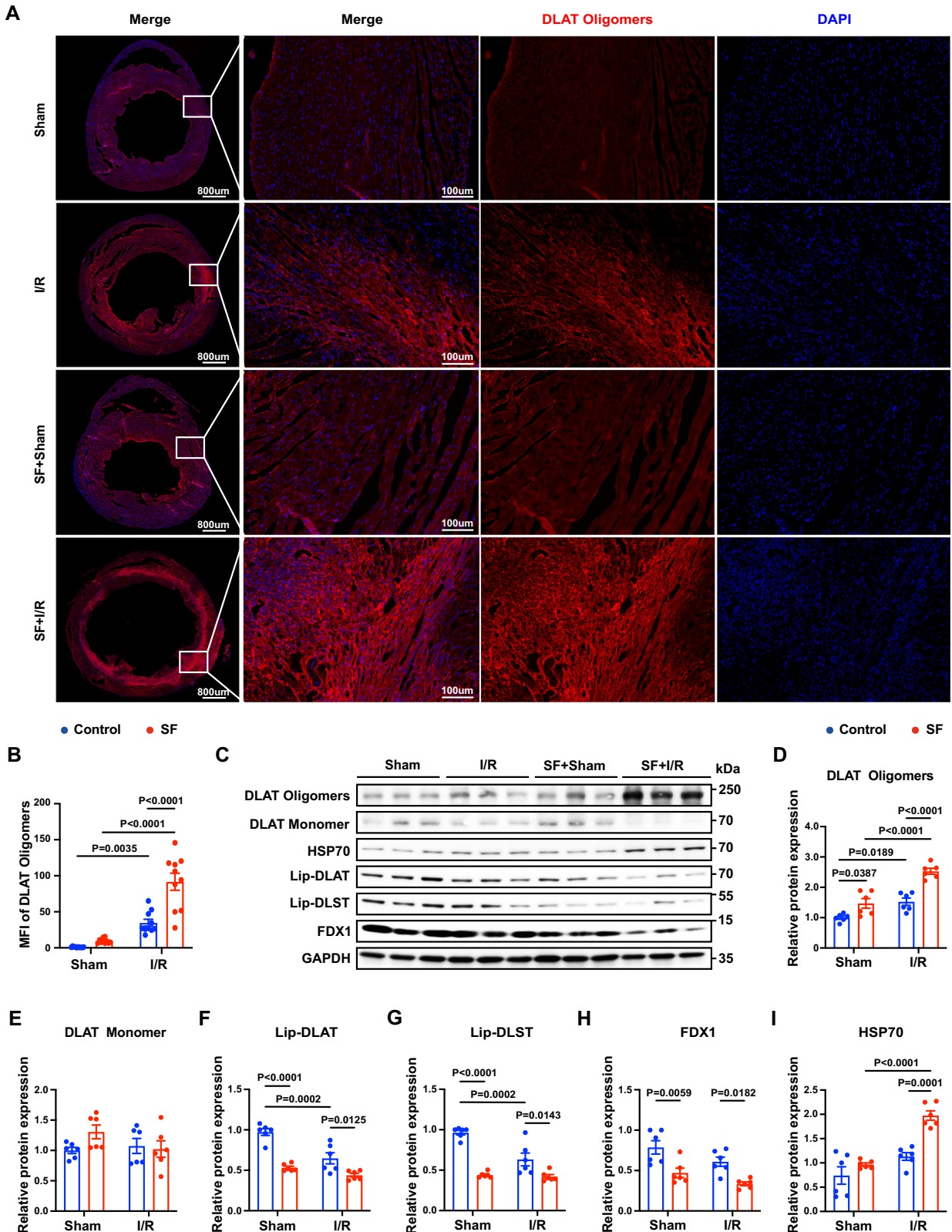

**Fig. 2 | Sleep fragmentation (SF) exacerbated cuproptosis after myocardial ischemia–reperfusion injury (MI/RI).** I/R, ischemia–reperfusion. **A** Representative images of DLAT oligomers from both wide-field and confocal immunofluorescence imaging of the anterior myocardial wall by confocal microscopy (red: DLAT oligomers, blue: DAPI). **B** DLAT oligomers were quantified by calculating the mean fluorescence intensity ($n = 10$ mice per group). **C** Validation of the levels of DLAT oligomerization, lipoylated proteins, iron-sulfur cluster proteins and HSP70 by Western blotting. **D**–**I** Statistical analysis of the levels of DLAT oligomers, DLAT monomer, Lip-DLAT, Lip-DLST, FDX1 and HSP70. Data on these proteins were normalized to GAPDH ($n = 6$ mice per group). Data are presented as the mean ± s.e.m. The significance of differences was evaluated using one-way ANOVA. Source data are provided as a Source Data file.

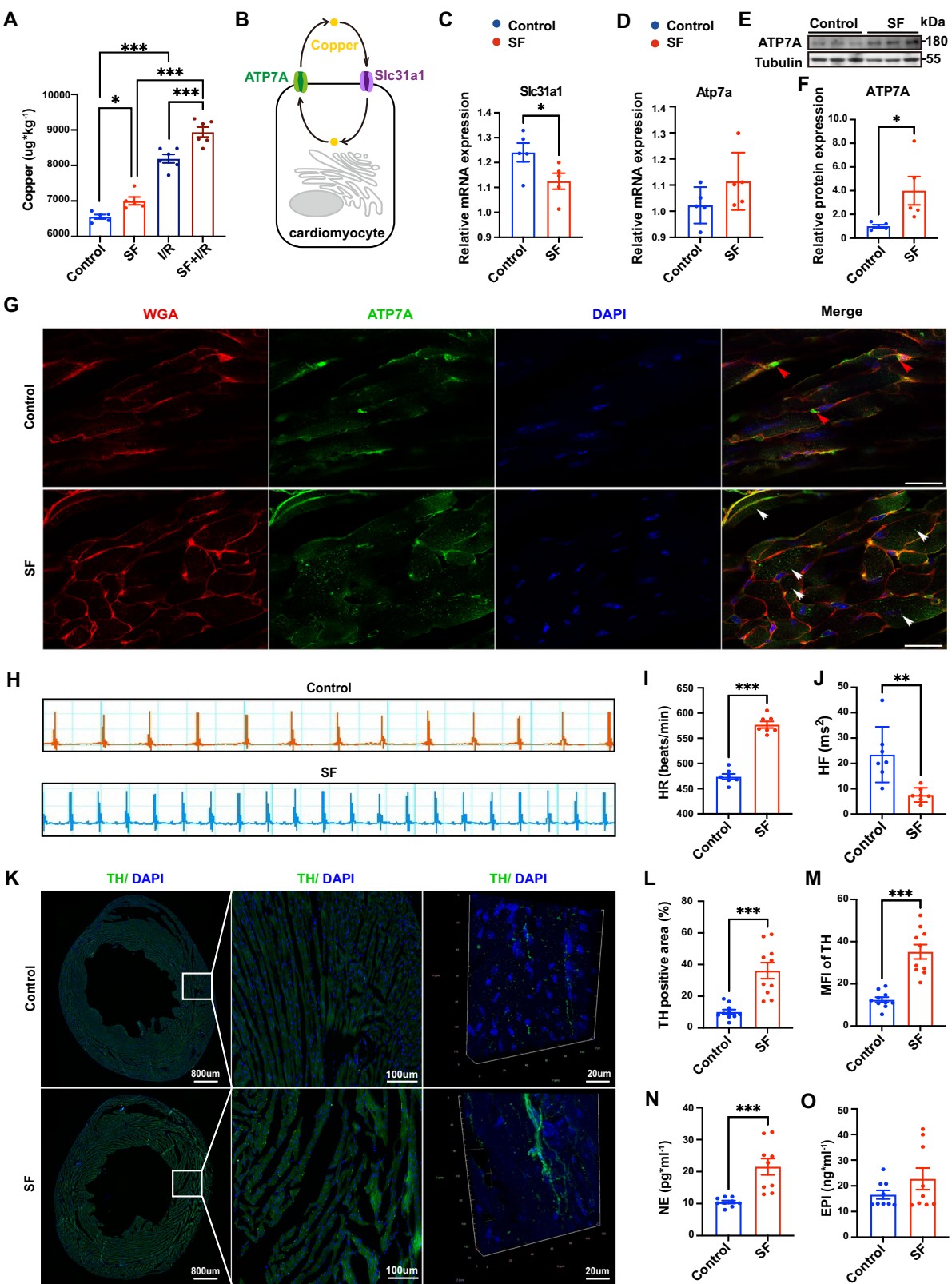

cell lines stably overexpressing VPS35 by lentiviral infection (Fig. 5D, E, S6A–S6E) and found that after NE treatment, VPS35-overexpressing cells (VPS35 OE) significantly improved intracellular copper transport and alleviated copper overload (Fig. 5F, G). Furthermore, VPS35 over-expression significantly reduced the oligomerization of DLAT (Fig. 5H–J, Fig. S7A, S7B), rescued the decrease in lipoylated protein levels (Fig. 5H, K, L) and the loss of iron-sulfur cluster proteins (Fig. 5H, M), and decreased the expression of HSP70 (Fig. 5H, N), indicating amelioration of cuproptosis and cell viability (Fig. 5O). These results indicated that sympathetic hyperactivity could impair intracellular copper transport, causing myocardium copper overload by inhibiting VPS35 expression.

**Fig. 3 | Fragmented sleep led to copper overload and increased sympathetic innervation in mice. A** Inductively coupled plasma–mass spectrometry (ICP–MS) revealed an increase in the content of myocardial copper ions in mice with sleep fragmentation (SF) and myocardial ischemia–reperfusion injury (MI/RI) ($n = 5, 5, 6,$ 6 for Control, SF, I/R, SF + I/R) Control vs. SF: $P = 0.0077$, the rest all $P < 0.0001$. I/R, ischemia–reperfusion. **B** Schematic diagram of copper transport in cardiomyocytes. **C, D** RT–qPCR analyses of the mRNA levels of Slc31a1 and Atp7a ($n = 5$ mice per group, $P = 0.0485$ for **C**, $P = 0.0550$ for **D**). **E** Validation of the expression levels of ATP7A in the myocardium by Western blotting. **F** Statistical analysis of ATP7A expression, which was normalized to tubulin ($n = 5$ mice per group, $P = 0.0372$). **G** Representative images of immunofluorescence staining of the myocardium by confocal microscopy (red, WGA; green, ATP7A; blue, DAPI) captured from the anterolateral wall of the mid-lower segment of the left ventricle. The red arrow indicates ATP7A located around the cell nucleus, while the white arrow indicates ATP7A free in the cytoplasm. Scale bar, 20 μm. **H–J** Cardiac electrophysiology

telemetry revealed that SF mice show faster resting heart rate (HR) and lower high frequency power spectral density (HF) ($n = 7$ mice per group, $P = < 0.0001$ for **I**, $P = 0.0030$ for **J**). **K** Representative images of sympathetic nerve terminals in mouse myocardial tissue slices were obtained by immunofluorescence staining of tyrosine hydroxylase (left: global image of the heart; middle: local image of the heart; right: 3D reconstruction of sympathetic nerves in thick slices of the heart). **L, M** The domination of myocardial sympathetic nerves was quantified by the percentage of TH-expressing area and the mean fluorescence intensity ($n = 10$ mice per group, $P = 0.0001$ for **L**, $P < 0.0001$ for **M**). **N, O** ELISA was performed to detect norepinephrine (NE) and epinephrine (EPI) in the cardiac tissue homogenate ($n = 8$ mice per group, $P = 0.0006$ for **N**, $P = 0.1908$ for **O**). Data are presented as the mean ± s.e.m. The significance of differences was evaluated using an unpaired two-tailed t test. $*P < 0.05$, $**P < 0.01$, $***P < 0.001$. Source data are provided as a Source Data file.

## Superior cervical ganglionectomy (SCGx) rescued myocardial copper overload and exacerbated MI/RI in mice with SF

In vivo, further investigation revealed increased NTF3, TH and C-Fos expression in the superior cervical ganglion (SCG) (Supplementary Fig. S3G–M). We hypothesized that the increased sympathetic activity in the heart in mice with SF could be attributed to enhanced signal transmission from the SCG. Therefore, we performed SCGx 20 min before MI/RI (Fig. 6A) and found that SCGx significantly rescued copper overload and improved MI/RI and myocardial apoptosis in mice with SF (Fig. 6B–H, Supplementary Fig. S8B–G). Mechanistically, SCGx restored the expression of VPS35 (Supplementary Fig. S8A, B) and ATP7A (Supplementary Fig. S8A, C) and eliminated the difference in myocardial copper levels between mice with SF and control mice (Fig. 6B). SCGx alleviated the altered oligomerization of DLAT (Fig. 6I–M), the decrease in lipoylated protein levels (Fig. 6K, N, O), the loss of iron-sulfur cluster proteins (Fig. 6K, P, Supplementary Fig. S8D–H), and the increase in HSP70 expression (Fig. 6K, Q) after MI/RI in mice with SF. Overall, due to its important role in the regulation of intracellular copper transport, sympathetic hyperactivity was proven to be an important causal link between sleep disorders and the exacerbation of heart damage.

## Copper chelation alleviates MI/RI exacerbation caused by sleep fragmentation in mice

To further validate the role of copper overload in MI/RI aggravated by SF. We utilized competitive copper chelator tetrathiomolybdate (TTM) to chelate copper ions, which has been extensively applied to explore the role of disrupted copper homeostasis in the onset and progression of multiple diseases[33,34]. In our study (Fig. 7A), treatment with TTM significantly blocked the upregulation of ATP7A (Supplementary Fig. S9A, B) and reduced myocardial copper levels in mice with SF (Fig. 7B). Compared with the solvent control, TTM pretreatment significantly improved cardiac function and reduced the infarct area and the myocardial apoptosis rate in mice with SF after MI/RI (Fig. 7C–H, Supplementary Fig. S9F–K). Furthermore, TTM also alleviated the decrease in lipoylated protein levels (Fig. 7K, N, O), the loss of iron-sulfur cluster proteins (Fig. 7K, P, Supplementary Fig. S9C–E), and the increase in HSP70 expression (Fig. 7K, Q). Finally, TTM administration inhibited the oligomerization of DLAT (Fig. 7I–M) and thus alleviated cuproptosis in mice with SF after MI/RI. These findings suggest that copper overload plays a key role in SF aggravated MI/RI.

## Discussion

Sleep disorders increase cardiovascular morbidity and mortality, and the purpose of this study was to investigate brain-heart interactions associated with sleep disorders. Our results suggest that sleep disruption promotes sympathetic overactivity, increases the germination of myocardial sympathetic nerve endings, and increases the level of NE in cardiac tissue, thereby inhibiting VPS35 expression and leading to

impaired copper transport and copper overload in cardiomyocytes. Copper overload further leads to exacerbate cuproptosis and apoptosis after myocardial infarction, and these effects can be rescued by excision of the sympathetic nerve or administration of a copper chelating agent. Our study elucidates a mechanism by which sleep disorders exacerbate myocardial damage and suggests possible targets for intervention.

Sympathetic dysfunction is a significant pathological alteration in sleep disorders[21]. Short-term sleep deprivation may lead to reduced sympathetic function in males[35,36], while chronic sleep deprivation can result in its hyperactivity[37,38]. The latter aligns with our findings of the effects of 16 weeks fragmented sleep. Cardiac sympathetic hyperactivation can increase the risk of developing hypertension[39,40], arrhythmias[41,42], endothelial dysfunction[43], and myocardial metabolic disorders[44,45]. In addition, the present study showed that sympathetic hyperactivity was responsible for cardiomyocyte copper metabolism disorder and exacerbated cuproptosis and apoptosis after MI/RI. Previous functional studies have confirmed the promoting effect of sleep deprivation on sympathetic nerve activity[46,47], which is consistent with our findings, and our study further revealed that anatomically, SF led to increased sprouting of myocardial sympathetic nerve terminals, which might be due to the influence of the central nervous system on SCG activity. This finding not only provides more conclusive anatomical evidence of brain-heart interactions but also proposes that cardiac sympathetic hyperactivity caused by sleep disorders is not only functional but also accompanied by organic changes, suggesting the significance of targeting increased sympathetic excitability for the treatment of insomnia.

Disruption of metal ion homeostasis plays a crucial role in the development of MI/RI[48]. Some studies have reported that ferroptosis is an important mode of injury after MI/RI[49–53]. In our study, no alterations in ferroptosis-related pathway molecules were detected 72 h after MI/RI; this is consistent with the results of another study, which found that there was a significant decrease in GPX4 expression at 24–48 h after MI/RI, but by 72 h, GPX4 expression showed a rebound and was not different from that of the controls[54]. In addition, we did not find differences in GPX4 expression after MI/RI or iron ion levels before MI/RI in the myocardium of mice with SF or control mice. Our results suggest that SF may not have a significant effect on ferroptosis and iron metabolism. Next, we explored copper metabolism and cuproptosis. A previous study reported that the levels of copper ions increased rapidly in the myocardium during I/R[55]. Copper overload can increase cellular oxidative stress[19,] activate apoptotic pathways[17–19], disrupt mitochondrial respiratory chain transmission[19], and trigger cellular cuproptosis[20]. Chronic copper overload has been reported to inhibit myocardial contractility by reducing calcium uptake in the sarcoplasmic reticulum[56]. Although there is some evidence of a correlation between copper metabolism and MI/RI[7,57,58], the present study verified the occurrence of myocardial cuproptosis in MI/RI.

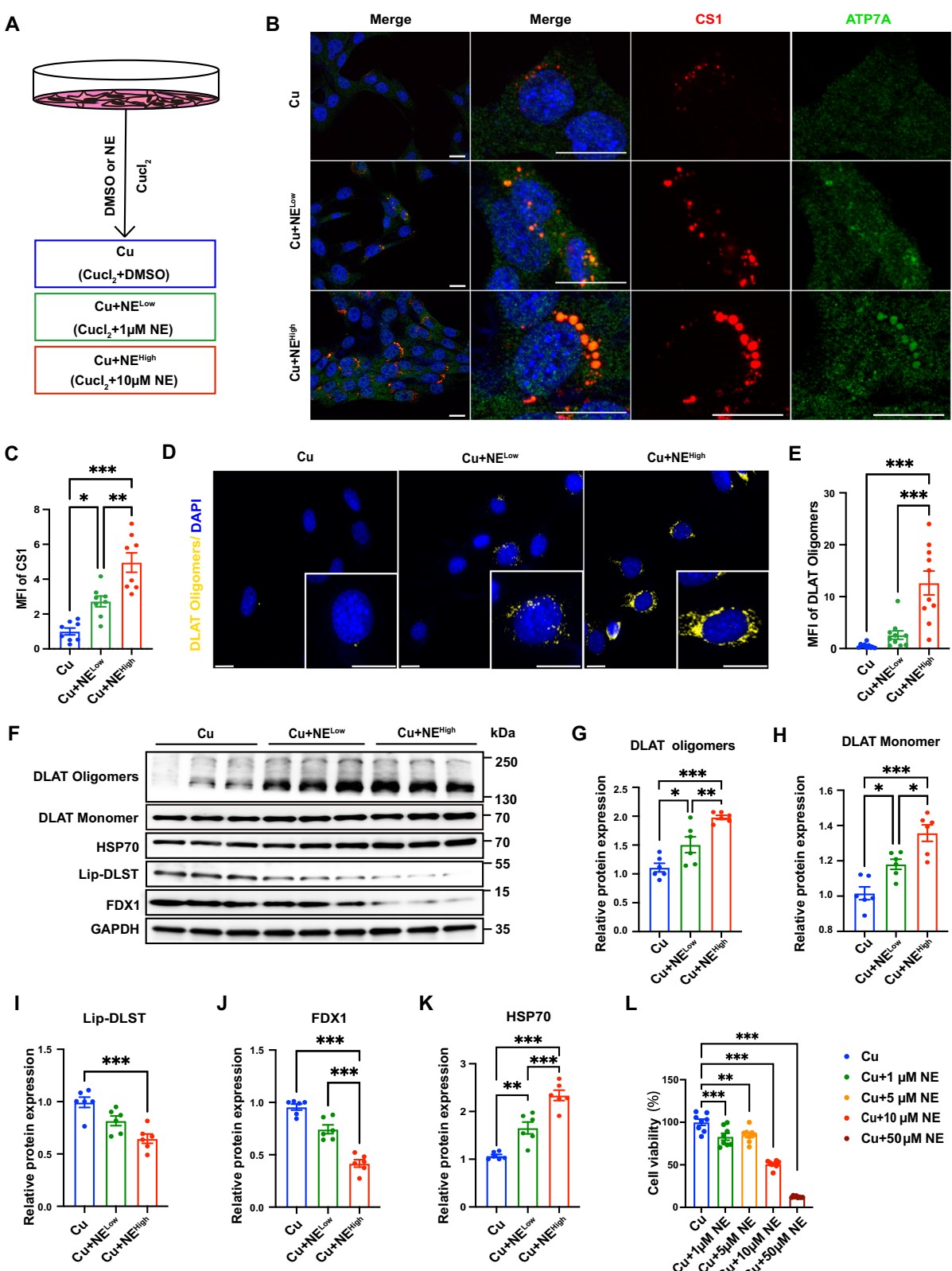

Furthermore, in our study, copper overload was proven to be the key causal link between sympathetic hyperactivation and aggravation of MI/RI in the SF mouse model. Interestingly, copper chelation treatment not only alleviated the exacerbating effect of SF on MI/RI but also ameliorated the damage caused by MI/RI itself, suggesting a promising strategy to ameliorate MI/RI.

Intraperitoneal administration of isoprenaline in rats led to changes in myocardial copper ion levels[59], indicating that ion homeostasis could be regulated by the autonomic nervous system. NE is the primary neurotransmitter released at sympathetic nerve terminals. In the present study, we not only successfully established an NE-induced copper overload model in cardiomyocytes in vitro but also

**Fig. 4 | Norepinephrine (NE) led to copper overload and cuproptosis in cardiomyocytes. A** Schematic diagram of the experimental groups in the cell model. HL-1 cells were treated with 10 μM CuCl$_2$ in the presence of solvent (DMSO), 1 μM NE or 10 μM NE for 24 h. **B** Representative images of copper overload and ATP7A localization in both wide-field and confocal immunofluorescence imaging by confocal microscopy (red: Coppersensor1 (CS1; indicates intracellular copper ions), green: ATP7A, blue: DAPI). Scale bar = 20 μm. **C** CS1 staining was quantified by calculating the mean fluorescence intensity ($n = 8$ independent experiments, Cu vs. Cu+NE^Low: $P = 0.0124$, Cu vs. Cu+NE^High: $P < 0.0001$, Cu+NE^Low vs. Cu+NE^High: $P = 0.0014$). **D** Representative images of DLAT oligomer immunofluorescence imaging by confocal microscopy. Scale bar = 20 μm. **E** DLAT oligomer levels were quantified by calculating the mean fluorescence intensity ($n = 10$ cells examined over 5 independent experiments, Cu vs. Cu+NE^Low: $P = 0.5537$, Cu vs. Cu+NE^High: $P < 0.0001$, Cu+NE^Low vs. Cu+NE^High: $P < 0.0001$). **F** Validation of the levels of DLAT oligomers, lipoylated proteins, iron-sulfur cluster proteins and HSP70 by Western blotting after treating HL-1 cells with 10 μM CuCl$_2$ in the presence of NE (1 μM, 10 μM) or DMSO for 24 h. **G–K** Statistical analysis of DLAT oligomers (Cu vs. Cu+NE^Low: $P = 0.0217$, Cu vs. Cu+NE^High: $P < 0.0001$, Cu+NE^Low vs. Cu+NE^High: $P = 0.0066$), DLAT monomer (Cu vs. Cu+NE^Low: $P = 0.0198$, Cu vs. Cu+NE^High: $P < 0.0001$, Cu+NE^Low vs. Cu+NE^High: $P = 0.0129$), Lip-DLST (Cu vs. Cu+NE^Low: $P = 0.0447$, Cu vs. Cu+NE^High: $P = 0.0003$, Cu+NE^Low vs. Cu+NE^High: $P = 0.0536$), FDX1 (Cu vs. Cu+NE^Low: $P = 0.0022$, Cu vs. Cu+NE^High: $P < 0.0001$, Cu+NE^Low vs. Cu+NE^High: $P < 0.0001$) and HSP70 (Cu vs. Cu+NE^Low: $P = 0.0018$, Cu vs. Cu+NE^High: $P < 0.0001$, Cu+NE^Low vs. Cu+NE^High: $P = 0.0005$). The data on these proteins were normalized to GAPDH ($n = 6$ independent experiments). **L** HL-1 cells were treated with 10 μM CuCl2 in the presence of different concentrations of NE for 12 h, and cell viability was determined using a CCK-8 assay ($n = 8$ independent experiments, Cu vs. Cu+1uM NE: $P = 0.0007$, Cu vs. Cu+5uM NE: $P = 0.0029$, Cu vs. Cu+10uM NE: $P < 0.0001$, Cu vs. Cu+50uM NE: $P < 0.0001$). All data are presented as the mean ± s.e.m. The significance of differences was evaluated using one-way ANOVA. *$P < 0.05$, **$P < 0.01$, ***$P < 0.001$. Source data are provided as a Source Data file.

successfully improved myocardial copper overload and cuproptosis after MI/RI in mice with SF by means of sympathetic ganglionectomy in vivo, indicating the key regulatory role of sympathetic nerve signals in intracellular copper ion metabolism. However, when investigating the causal relationship between sympathetic signaling and cardiac cuproptosis through SCGx in vivo, a contradictory finding arises in the expression changes of DLAT oligomers. SCGx in SF mice significantly alleviates the increase of DLAT oligomers after MI/RI, whereas the same effect is not observed in mice with normal sleep. We believe that this difference is mainly due to the 16 weeks of chronic copper accumulation in SF mice. Therefore, the increase in DLAT oligomers in the SF + IR group are a comprehensive effect of acute and chronic copper accumulation, with a higher baseline copper load than Sham+IR mice. The experimental data also indicate that prolonged chronic copper accumulation may increase the expression of copper transport proteins, including ATP7A, in cardiomyocytes. When SCGx normalizes copper efflux in cardiomyocytes, the efficiency of copper ion efflux should also be higher in cardiomyocytes of SF mice. Therefore, SCGx-induced alleviation in cuproptosis is more pronounced in the SF + IR mouse than in the control group, whether from the perspective of baseline copper load or improved efficiency of copper ion efflux. Additionally, to ensure the consistency of the results detection time, copper ion detection 3 days after SCGx shows a continuous efflux of intracellular copper to lower levels in both groups. However, the peak difference in the formation of DLAT oligomers induced by MI/RI may have occurred at an earlier time point, which may also lead to inconsistency in the trends of copper ion levels and DLAT oligomer levels.

Since sympathetic signaling can modulate copper content, we further explored the molecular mechanism by which sympathetic signals regulate copper transport in cardiomyocytes. The normal functioning of myocardial vesicle trafficking is necessary for copper ion efflux. As reported previously[25,26,60], the copper ion transporter ATP7A carries copper ions out of the trans-Golgi network (TGN) and forms copper-transporting vesicles; this is the first step of copper efflux from the cell. Then, ATP7A is transported outward, and the vesicles fuse with the plasma membrane to release the copper ions. Finally, ATP7A is endocytosed to form endosomes and returns to the TGN for the next round of cycling with the help of the retromer complex. Complete copper ion efflux requires the retrieval and recycling of ATP7A between the TGN and the plasma membrane. Therefore, in this study, we utilized ATP7A as an important tracer protein and used a CS1 copper ion probe to reveal the mechanism of intracellular copper overload induced by NE. In both the myocardium of mice with SF and HL-1 cells treated with NE, we found that ATP7A was blocked in vesicles, and most of these vesicles were colocalized with copper ions, suggesting that ATP7A carrying copper ions might be defective in membrane fusion, causing stagnation of the copper transport system and resulting in intracellular copper overload and even cuproptosis.

Intracellular copper transport is a complex multimolecular and collaborative biological process involving multiple proteins. ATOX1 is responsible for transferring intracellular copper ions to ATP7A to form copper transport vesicles[61,62]. AKT2 and VPS35 play a key role in the stabilization and translocation of ATP7A protein to the plasma membrane[63]. PTBP regulates the excretion of copper ions by regulating the expression of copper transporters[64], and LOX1 promotes its entry into the secretion pathway after binding with copper ions[65]. VPS18 has been reported to be an important molecule in vesicle translocation to lysosomes[30]. SNX12, SNX27 and VPS35 may play decisive roles in the endocytosis and retrograde transport of ATP7A back to the TGN[31,32]. COMMD1 is intricately involved in the copper-dependent trafficking of ATP7A between the trans-Golgi network and vesicles in the cellular periphery[66]. AP-1, COG1 and VPS35 are responsible for screening out ATP7A/B in endosomes for recycling to the TGN[60,67,68]. Among these proteins, VPS35, a core component of the retromer complex, which might play an important role in both the efflux of copper ions from ATP7A-containing vesicles and the recycling of ATP7A from endosomes to the TGN[32,68], was downregulated both in the myocardium of mice with SF and HL-1 cells treated with NE. It has been shown that knockdown of VPS35 prevents ATP7A from fusing with the plasma membrane and becoming blocked in cytoplasmic vesicles[32], which is consistent with the phenotype observed in the present study. Our research further proved that overexpression of VPS35 could significantly relieve the stagnation of ATP7A and rescue copper overload and cuproptosis in NE-treated HL-1 cells. In addition, another important role of VPS35 is to participate in the construction of the retromer complex for retrieval and recycling of cargo from endosomes back to the TGN. Deletion of VPS35 results in the inability of a variety of cargoes to return from endosomes to the TGN, including ATP7B[68], a homologous complex of ATP7A, but whether the return of ATP7A is also blocked remains to be further elucidated. In conclusion, the inhibition of VPS35 expression by NE treatment caused copper overload in HL-1 cells mainly through disruption of fusion between copper-transporting vesicles and the plasma membrane. Decreased VPS35 expression might also lead to the obstruction of ATP7A recycling and aggravate the disorder of copper transport.

The stress- or hypoxia-associated molecular and biological changes accompanying MI/RI may represent crucial avenues for future exploration into the mechanisms underlying cardiac copper overload. Although there is limited research on the regulation of VPS35 expression in cardiomyocytes, cutting-edge studies in the field of oncology have provided some potential insights. Some articles suggest that inhibiting the function of the heat shock protein 90 (Hsp90) can promote the expression of VPS35 by upregulating Bclaf-1, thereby facilitating extracellular vesicle release in liver cancer cells[69]. Since Hsp90 upregulation was observed in response to cellular stress or

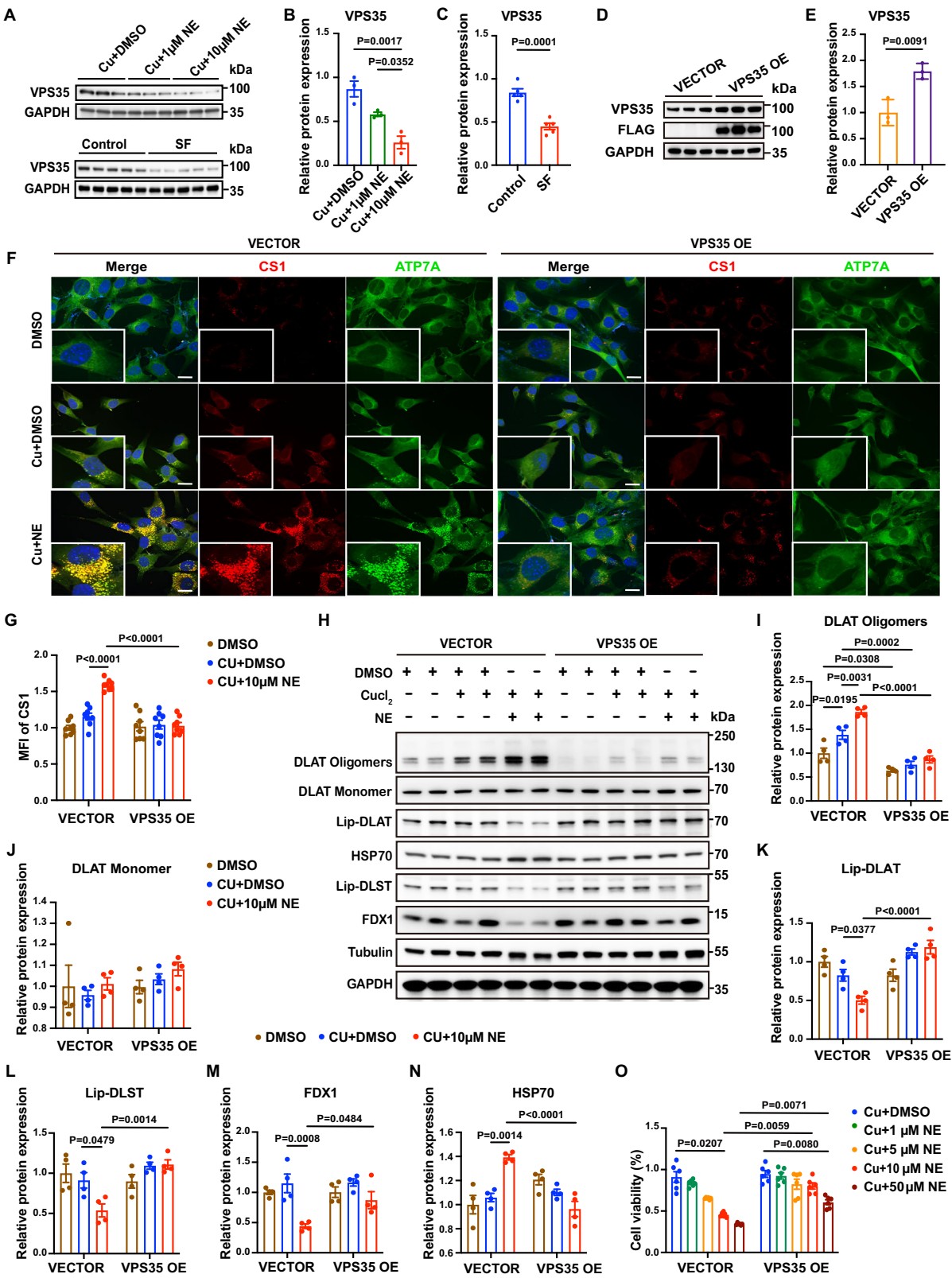

hypoxia signals[70,71], sleep disorders or IR injury may mediate copper overload in myocardial cells through the Hsp90-Bclaf1-Vps35 pathway, providing a crucial research direction for our subsequent investigations into the underlying mechanisms. In addition to VPS35, which is a critical regulatory protein governing intracellular copper transport, COMMD1 holds potential research value in the investigation of cardiac copper overload. The studies of James Kang's group reported

increased copper efflux from the heart to the peripheral blood associated with COMMD1 upregulation, and knockout of COMMD1 was found to significantly reduce the infarct size and effectively preserved myocardial contractile function in mice subjected to LAD coronary artery ligation[72]. The authors suggested that the cardioprotective effect of COMMD1 deletion involved the alleviation of copper loss. However, consistent with our experimental data, numerous studies

**Fig. 5 | Overexpression of VPS35 rescued copper overload and cuproptosis caused by norepinephrine (NE). A** Validation of the expression levels of VPS35. Statistical analysis of VPS35 expression ($n = 3$, independent experiments for **B**; $n = 5$ mice for **C**). **D** Validation of the overexpression of VPS35 and Flag in HL-1 cells 96 h after transfection with the lentivirus. **E** Statistical analysis of VPS35 overexpression (OE) ($n = 3$ independent experiments, $P = 0.0091$). **F** Representative images of copper overload and ATP7A localization in both wide-field and confocal immunofluorescence images by confocal microscopy (red: CS1, green: ATP7A, blue: DAPI). **G** CS1 staining was quantified by calculating the mean fluorescence intensity ($n = 8$ independent experiments). **H** Validation of the levels of DLAT oligomerization, lipoylated proteins, iron-sulfur cluster proteins and HSP70 by Western blotting after treating vector- and VPS35-overexpressing HL-1 cells with 10 μM $CuCl_2$ and 10 μM NE for 24 h. **I–N** Statistical analysis of the levels of DLAT oligomers, DLAT monomer, Lip-DLAT, Lip-DLST, FDX1 and HSP70. The data on Lip-DLAT and Lip-DLST were normalized to tubulin, and the other data were normalized to GAPDH ($n = 4$ independent experiments). **O** Vector- and VPS35-overexpressing HL-1 cells were treated with 10 μM CuCl2 in the presence of different concentrations of NE for 12 h, and cell viability was determined using a CCK-8 assay ($n = 6$ independent experiments). All data are presented as the mean ± s.e.m. The significance of differences was evaluated using one-way ANOVA for **B**, **G** and **I–N**, unpaired two-tailed t-test for **C** and **E**, and two-way ANOVA for **O**. Source data are provided as a Source Data file.

have indicated the efficacy of copper chelators in improving the prognosis of ischemic heart disease in mice, contradicting the hypothesis proposed by James Kang. Therefore, we propose that in the early stages of ischemic heart disease, the predominant cellular damage effects in the early stages of injury are primarily attributed to the copper overload in heart, which activate COMMD1 transcription to facilitate compensatory copper efflux. However, the upregulation of COMMD1 may have dual effects: on the one hand, COMMD1 can inhibit HIF-1 transcription[73,74], thereby further participating in the activation of signaling pathways related to the deterioration of heart function. On the other hand, it may regulate intracellular copper content by modulating the stability and function of ATP7A and ATP7B[75]. Therefore, the cardioprotective effect of COMMD1 deletion may need to be comprehensively reconsidered from the perspective of both copper-related and non-copper-related processes. Exploring the mechanisms underlying copper overload remains a pivotal direction for future research in the field of copper-related cardiac injuries.

The major limitations of this study are as follows. First, only male mice were used for the study. Although sex plays an importance in cardiac injury during sleep disorders, the aim of this study was to investigate how the sympathetic nervous system regulates copper metabolism, and female mice typically exhibit attenuated MI/RI and exacerbated injury following sleep deprivation. Future studies are warranted to investigate the differential effects of sleep disorders on cardiac injury in female and male mice. Second, Copper overload has been shown to be a crucial intermediate in the exacerbation of myocardial damage by sleep disorders. However, due to ethical constraints, our study did not go on to validate impaired copper transport in the myocardium of patients with sleep disorders, which will require the development of non-invasive means to validate in the future. Third, the causality between VPS35 inhibition and copper metabolism disorders was partially demonstrated in the present study, however VPS35 was not knocked out in vivo to further confirm the relationship between VPS35 and copper overload, as well as the upstream regulatory mechanism in brain-heart comorbidities. Last, the findings tentatively suggest that copper overload may be directly involved in the progression after MI/RI, but we did not further investigate the role of VPS35 inhibition and copper overload in ischemic cardiomyopathy or set up in vivo experiments to validate it, which will be the main direction of investigation in our future work.

In summary, we describe the brain-heart crosstalk mechanism in which SF increases sympathetic hyperactivity and then triggers cardiomyocyte copper overload, while exacerbating myocardial apoptosis and cuproptosis after MI/RI. Our data identify copper metabolism as a sensor in the heart; Signals are received from the brain and sympathetic nerves, and copper metabolism acts as a regulator to control the occurrence of downstream cuproptosis and apoptosis. The cardiac copper transport signals are a potential therapeutic target in sleep disorder-related myocardial injury and even in ischemic myocardial injury.

## Methods

### Animals
Adult male C57BL/6 mice (6–8 weeks old, weighing 21–23 g; male) were purchased from Hunan SJA Laboratory. All mice were raised in standard plastic cages at a temperature of 20–26 degrees Celsius and a humidity of 50–60%, with a 12-h light/dark cycle, and had libitum access to food and water. All experimental procedures and protocols were approved by the Xiangya Hospital experimental animal Ethics Committee of Central South University (permit code: 2021111240) and adhered to the National Institute of Health Guide on the Care and Use of Laboratory Animals. Randomization and blinding were used. Briefly, mice were randomly assigned to experimental and control groups after being tagged with animal ear tags. Echocardiographic measurements and modeling of cardiac ischemia-reperfusion injury were performed by an experienced operator in a blinded fashion. Throughout the experiment, the operator responsible for the experimental procedure and data analysis was unaware of the group assignment.

### Mouse model of sleep fragmentation (SF)
As in previous study[76], mice were placed in a sleep deprivation chamber (Anhui Yaokun Biotechnology, ZL-013). The sweep bar moved along the bottom of the cage every 2 min during the light cycle (ZT0–12) and was stationary during the dark cycle (ZT12–24) for 16 weeks. Control mice that received undisturbed sleep were placed in sleep fragmentation chambers with stationary sweep bars.

### Animal anesthesia
All mice if needed were anaesthetized by inhalation of 4–6% (v/v) sevoflurane (Hengrui, China), and anesthesia was maintained with 2% (v/v) sevoflurane. For anesthesia that requires interruption of breathing, such as myocardial ischemia-reperfusion modeling, tracheal intubation was performed with a 20 G catheter, and machine-controlled breathing was performed with a rodent ventilator (Harvard Apparatus, 55-0000) at an inspiratory to expiratory ratio of 1:1.5 and a frequency of 120 breaths/minute.

### Myocardial ischemia-reperfusion injury (MI/RI) model
To induce MI/RI model, mice were subjected to left anterior descending coronary artery ligation (LAD) as described previously[77]. Briefly, the animals were anesthetized, intubated, and ventilated. The heart was exposed through a left-sided minithoracotomy between the fourth and fifth ribs, and 10-0 prolene suture (NINGBO MEDICAL, LingQiao) was passed underneath the LAD at 2–3 mm distal to its origin between the left auricle and conus arteriosus. A suture loop around the artery was tightened to ensure the LAD occlusion. Regional ischemia was confirmed by the elevation of the S-T segment on the electrocardiogram (ECG) and visual inspection of discoloration of the occluded distal myocardium. After 30 min of ischemia, the ligation was released, and the heart allowed to be repercussed as confirmed by visual inspection. Then, the animals were allowed to awaken on a warm pad and kept in a cage with free access to wet food and water during

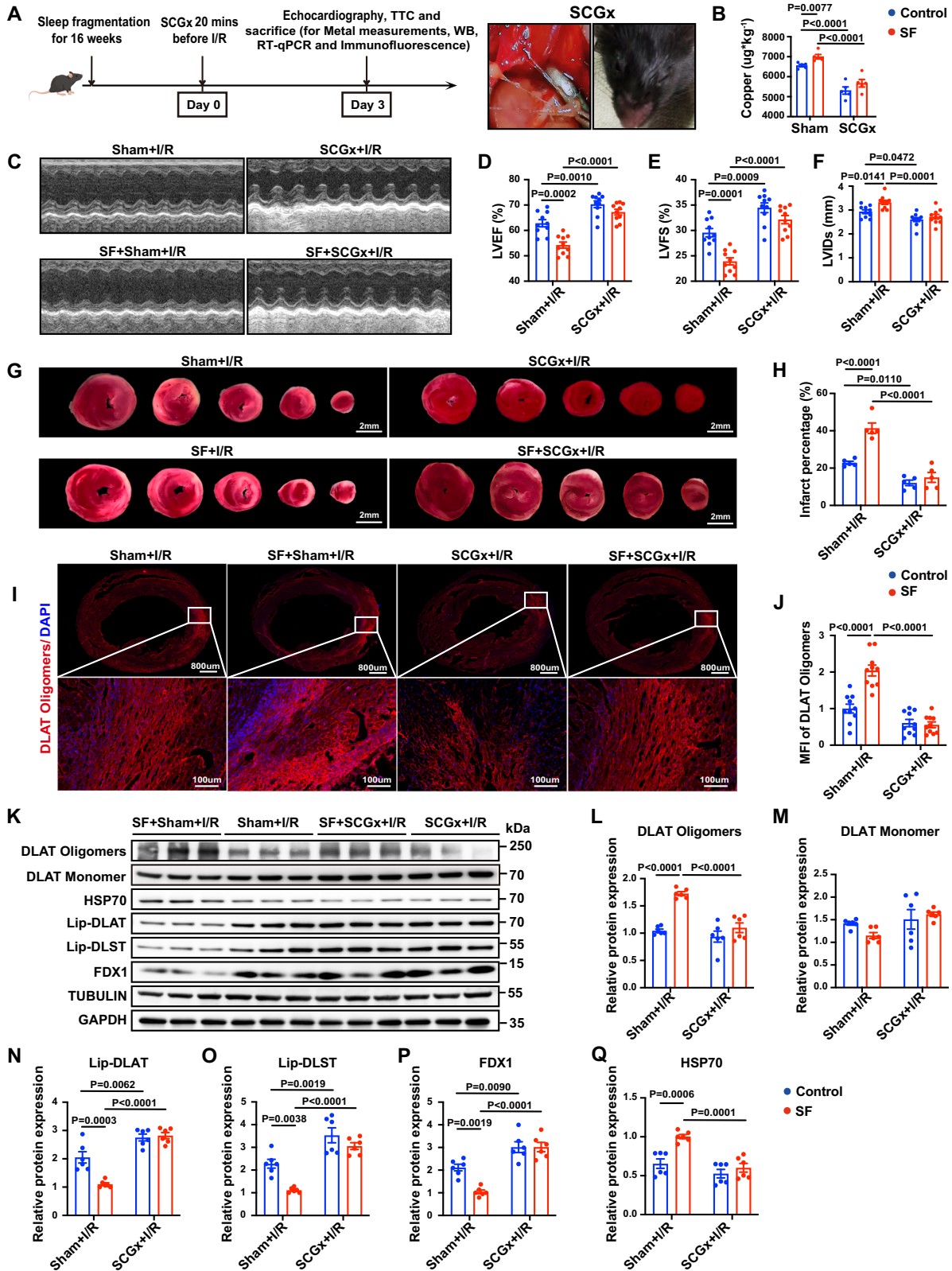

the reperfusion period. The mice in Sham group only underwent the same surgical procedure except for the ligation of LAD.

**Superior cervical ganglionectomy model (SCGx)**
SCGx was performed as described previously[78]. After induction of anesthesia and intubation, a vertical neck incision was made. Lateral positioning of the sternocleidomastoid muscle allowed for clear

visibility of the carotid artery. The almond shaped SCG is located behind the carotid bifurcation. And the ganglionic cell body was fully extracted from the sympathetic chain by shifting the external carotid artery to the lateral side (Fig. 6A) 20 min before MI/RI in the SCGx group. In the sham-operated group, only the skin incision and SCG isolation were performed. The SCG tissue was collected to analyze the reasons for the increased sprouting of sympathetic nerve terminals.

**Fig. 6 | Superior cervical ganglionectomy (SCGx) attenuated MI/RI and cuproptosis in mice with sleep fragmentation (SF). A** Schematic of the experimental protocol used to establish SCGx-exacerbated MI/RI (left). Removal of the SCG ganglion from the posterior aspect of the common carotid artery was performed under a stereomicroscope, and images of mice that developed ipsilateral ptosis after the SCG was removed from one side were captured. Cartoon mouse images was drawn by Figdraw (https://www.figdraw.com/#/)**. B** SCGx reduced the copper ion level in the myocardium in mice with SF, as detected by ICP–MS (*n* = 5 mice per group). Unpaired two-tailed mutiple t-test was used. **C** Representative M-mode echocardiographic changes showed that SCGx attenuated the impaired cardiac function in mice with SF and MI/RI. **D–F** Statistical analysis of Left ventricular ejection fraction (LVEF), Left ventricular fractional shortening (LVFS) and Left ventricular internal diameter, systolic (LVIDs) (*n* = 10 mice per group).

**G** Representative TTC staining of the myocardium. Scale bar, 2 mm. **H** The percentage of infarct area was determined by TTC staining (*n* = 5 mice per group). **I, J** Representative images of DLAT oligomers from both wide-field and confocal immunofluorescence images of the myocardium by confocal microscopy (red: DLAT oligomers, blue: DAPI). DLAT oligomers levels were quantified by calculating the mean fluorescence intensity (*n* = 10 mice per group). **K** Validation of the levels of DLAT oligomerization, lipoylated proteins, iron-sulfur cluster proteins and HSP70 by Western blotting. **L–Q** Statistical analysis of DLAT oligomers, DLAT monomer, Lip-DLAT, Lip-DLST, FDX1 and HSP70. Data on DLAT oligomers were normalized to tubulin, and the remaining data were normalized to GAPDH (*n* = 6 mice per group). Data are presented as the mean ± s.e.m. The significance of differences was evaluated using one-way ANOVA. Source data are provided as a Source Data file.

## Tetrathiomolybdate (TTM) treatment

After the mice were subjected to 16 weeks of SF intervention or not, the SF and control mice were randomly divided into 2 groups respectively, while the mice in the TTM group were given TTM 80 mg/kg by daily gavage for two weeks before MI/RI, and the other group was given the corresponding solvent gavage manipulation.

## Measurement of metal ions

Metal ion levels in myocardial tissues were detected as before[79]. In short, first, we weighed approximately 0.1 g of cardiac tissue taken from the entire left ventricle below the ligature suture plane and placed it in a 15 ml centrifuge tube. Add 1.2 ml of 68% HNO3 and heated at 280 °C for 120 min. Then, add 400 μl of 30% H2O2 and digested the tissue until it becomes clear and transparent. Levels of major physiological metal ions were measured by Inductively Coupled Plasma – Mass Spectrometry (ICP-MS) (Perkin Elmer NexION 2000).

## Resting Heart Rate (HR) examination

A telemetry probe was implanted into the subcutaneous tissue of a mouse to monitor its resting HR. First anesthetizing, then placing a sterilized telemetry probe in the subcutaneous tissue on the back of the mouse, with electrodes fixed behind the sternocleidomastoid muscle in the suprasternal notch and the xiphoid process. Finally, the skin was sutured, and the mouse was placed freely on a plate that could be detected by the telemetry system (AD Instruments, MT 10B). Data was recorded using the LabChart software (version 8.1.24, Australia). Two to ten minutes electrocardiogram segments were recorded and analyzed by heart rate variability (HRV) module of LabChart. Spectrum was integrated in low-frequency (LF, 0.20–0.75 Hz) and high-frequency (HF, 0.75–2.5 Hz) bands. The HF was to evaluate cardiac sympathovagal balance as previously reported[80,81].

## Echocardiographic measurements

The cardiac function collects and was analyzed at days 0, 3 and 7 after MI/RI using transthoracic echocardiography (Shenzhen Mindray Biomedical Electronics, M9CV) as previously described[82,83]. In brief, mice were anesthetized with a HR at 400–500 beats/min, gently restrained and placed on a platform. Hair removal cream was then used to remove hair from the anterior chest. Medical ultrasound gel was used as a coupling agent between the ultrasound scanning head and the skin. M-mode echocardiography was performed with L16-4Hs (4–16 MHz) transducer at the level of the papillary muscle to measure left ventricular ejection fraction (LVEF), left ventricular fractional shortening (LVFS), left ventricular internal diameter systolic (LVIDs), left ventricular internal diameter diastolic (LVIDd), left ventricular end diastolic volume (LVEDV) and left ventricular end systolic volume (LVESV). LVEF was calculated as LVEF = (LVEDV − LVESV)/LVEDV × 100%, and LVFS was calculated as LVFS = (LVIDd − LVIDs)/LVIDd × 100% by Microsoft (Version 16.8) and GraphPad Prism software (Version 9.2.0).

## TTC staining

To measure the infarct size, hearts were collected at 3 days after MI/RI. After being excised and rinsed in phosphate buffer saline (PBS) for washing out remaining blood, the heart was frozen at −20 °C for 15 min and cut horizontally into slices (5 slices/heart). Then, the slices were incubated for 20 min with 1% 2,3,5-triphenyltetra-zolium chloride (TTC, Sigma, T8877) at 37 °C followed by 4% paraformaldehyde fixation for 24 h to visualize the unstained infarcted region. The infarct area was expected to be stainless, the normal tissue to be stained in red. The left ventricle infarct size was analyzed using ImageJ (1.53a) analysis software and expressed as the mean percentage of infarct area to the total area[77].

## Cardiac tissue preparation

Animals were anaesthetized, and humanely sacrificed by cervical dislocation and decapitation. The left ventricular cardiac tissues harvested from mice were fixed with 4% paraformaldehyde, embedded in optimal cutting temperature compound (SAKURA, 4583) frozen section, and cut at 5 μm with a microtome. For the molecular biological analysis, including Western blot analysis and qRT-PCR analysis, the samples were taken from the ischemic area and a small portion of the marginal tissue. These samples were obtained from the left ventricle below the ligature. After being removed from the mice, the samples were immediately washed to remove any remaining blood using ice-cold saline. Subsequently, they were dissected and stored at −80 °C for further analysis.

## Cell culture

Murine atrial myocytes, HL-1, were purchased from Changsha Abiowell Biotechnology Co., Ltd. The cells were cultured with MEM supplemented with 10% FBS, 100 U/ml penicillin/streptomycin. Cells were all grown at 37 °C in a humid atmosphere of 5%CO2 and 95% air. Norepinephrine (NE, MCE, HY-13715), Phenelzine (MCE, HY-B1018A) and Cucl2 (Sigma, C3279) for treatment of cells were dissolved in DMSO, DMSO and DEPC water respectively, and then diluted with culture medium.

## Construction of VPS35 overexpression HL-1 cell line

Lentiviral vectors containing the musVps35-Flag (VPS35 OE) sequences and the corresponding control lentivirus vector (VETOR) were constructed by HonorGene (Changsha, China). The titer of VPS35 OE was 1.5 × 10^8 TU/ml, and the titer of VECTOR was 4 × 10^8 TU/ml. HL-1 cells were transfected with the Co-incubation of transfection reagent GA (Genechem, GRCT101B) and lentivirus at an MOI of 100. Green fluorescent protein was detected in HL-1 cells using a fluorescence microscope (Nikon, Japan) to confirm successful transfection. Puromycin (MCE, HY-15695) for screening and purification of the transfected cells. The overexpression efficiencies of VPS35 were identified using qRT−PCR and Western blotting.

## Immunofluorescence staining

Histological sections were air-dried, washed off OCT, blocked with 5% goats' serum for 60 min in room temperature, then incubated with

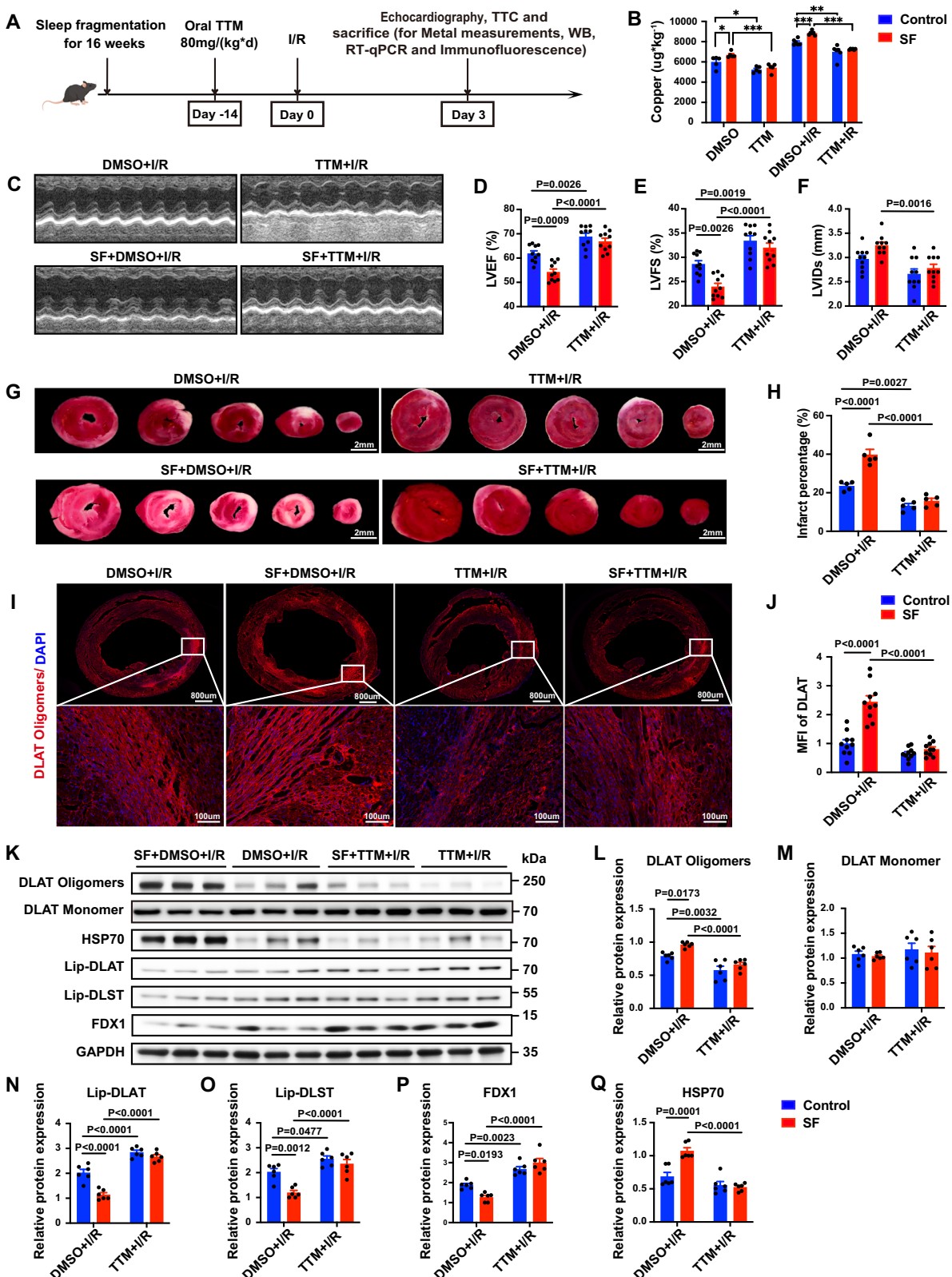

antibody against DLAT (SANTA, sc-271534), TH (Abcam, ab137869) or ATP7A (SANTA, sc-271534) at 4 °C overnight and incubated with secondary antibody (Alexa Fluor 488 or Fluor 594) for 1 h at room temperature. WGA staining involves incubating sections with WGA at 37 °C for 40 min before blocking, followed by the same steps as described above. For cell immunofluorescence staining, cells are fixed with 4% PFA for 10 min before blocking, followed by the same steps as tissue

section staining. To obtain the staining results of DLAT and TH for the global scanning of the heart, scans were performed with a 20x objective lens using a fully automatic multimarker multispectral quantitative pathology imaging system (AKOYA BIOSCIENCES) equipped with Vectra Polaris software after the staining and mounting. Upright microscopes (Nikon N2Ti2-A) are used for brightfield imaging. The three-dimensional reconstruction of the myocardium stained with TH

**Fig. 7 | TTM attenuated MI/RI and cuproptosis in mice with sleep fragmentation (SF). A** Schematic of the experimental protocol used to establish TTM-exacerbated MI/RI. Cartoon mouse images was drawn by Figdraw (https://www.figdraw.com/#/). **B** TTM reduced the copper ion level in the myocardium in SF mice before and after MI/RI, as detected by ICP–MS ($n$ = 5, 5, 6, 6 mice for DMSO, TTM, DMSO + I/R and TTM + I/R). The significance of differences was evaluated using an unpaired two-tailed multiple t-tests. *$P < 0.05$, **$P < 0.01$, ***$P < 0.001$. **C** Representative M-mode echocardiographic changes. **D–F** Statistical analysis of Left ventricular ejection fraction (LVEF), Left ventricular fractional shortening (LVFS) and Left ventricular internal diameter, systolic (LVIDs) ($n$ = 10 mice per group). **G** Representative TTC staining of the myocardium. Scale bar, 2 mm. **H** The percentage of infarct area was determined by TTC staining ($n$ = 5 mice per group). **I, J** Representative images of DLAT oligomers from both wide-field and confocal immunofluorescence images of the heart by confocal microscopy (red: DLAT oligomers, blue: DAPI), and the levels of DLAT oligomers quantified by calculating the mean fluorescence intensity ($n$ = 10 mice per group). **K** Validation of the levels of DLAT oligomerization, lipoylated proteins, iron-sulfur cluster proteins and HSP70 by Western blotting. **L–Q** Statistical analysis of DLAT oligomers, DLAT monomer, Lip-DLAT, Lip-DLST, FDX1 and HSP70. Data on these proteins were normalized to GAPDH ($n$ = 6 mice per group). Data are presented as the mean ± s.e.m. The significance of differences was evaluated using one-way ANOVA. Source data are provided as a Source Data file.

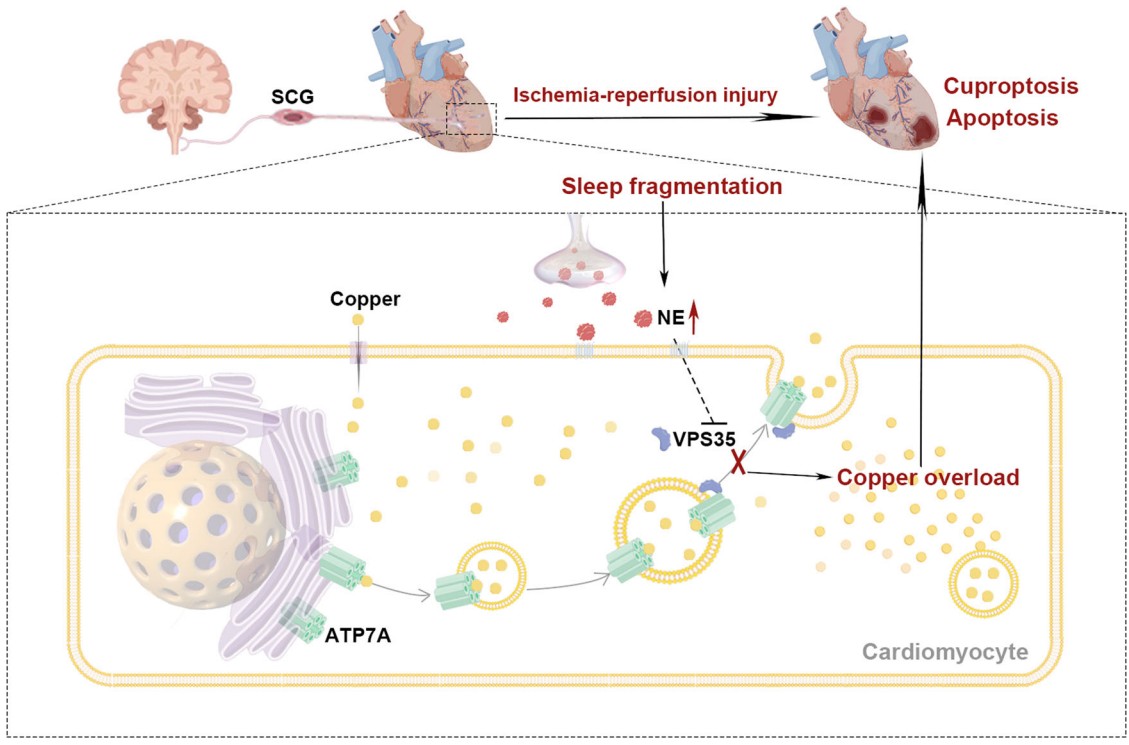

**Fig. 8 | Model of sleep fragmentation exacerbates myocardial ischemia–reperfusion injury by promoting copper overload in cardiomyocytes.** Sleep fragmentation increases sympathetic innervation of the myocardium via SCG, releasing more norepinephrine, which indirectly inhibits VPS35 in cardiomyocytes, leading to ATP7A-associated impairment of copper transport, triggering intracellular copper overload and ultimately exacerbating myocardial ischemia-reperfusion injury. The figure was drawn by Figdraw (https://www.figdraw.com/#/) and Adobe Photoshop.

was obtained by scanning with a 40x objective lens using a confocal microscope (ZEISS, LSM900) equipped with ZEN software (Carl Zeiss, v2009) and then performing 3D rendering. The fluorescence results of other cells for ATP7A and DLAT were acquired with a 63× oil immersion lens using a microscope (ZEISS ApoTome.2) equipped with ZEN software (Carl Zeiss, v2009). Imagej (1.53a) and Fiji (v2.9.0) were used to process, analyze and count immunofluorescence pictures.

### Coppersensor-1 (CS1)
CS1 (Psaitong, C11621)[29,84], a synthetic fluorophore for live-cell copper imaging, is a small-molecule, membrane-permeable fluorescent dye for imaging labile copper pools in biological samples, including live cells. This probe, comprising a boron dipyrromethene chromophore coupled to a thioether-rich receptor, has a picomolar affinity for Cu+ with high selectivity over competing cellular metal ions. CS1 fluorescence increases up to 10-fold on binding to Cu+. The staining involves replacing the cell culture medium with 5 µM CS1 and then incubating the cells in a dark environment at 37 °C for 20 min before fixation and subsequent staining. The cardiac tissues and cell were counter stained with 4′,6-diamidino-2-phenylindole (DAPI, Sigma). The fluorescence

results of CS-1 were acquired with a 63× oil immersion lens using a microscope (ZEISS ApoTome.2) equipped with ZEN software (Carl Zeiss, v2009).

### TUNEL staining
The heart samples were separated and fixed in 10% phosphate-buffered formalin for 24 h, subsequently embedded in paraffin, sliced (4 µm). Terminal deoxynucleotidyl transferase-mediated dexoyuridine triphosphate nick-end labeling (TUNEL) staining was performed using the TUNEL BrightGreen Apoptosis Detection Kit (Roche Life Science, A112-01) following manufacturer's instructions. Apoptotic nuclei were labeled with green fluorescein staining and total cardiomyocyte nuclei were marked with 4′,6-diamidino-2-phenylindole (DAPI) (SouthernBiotech, 0100-20). The slices of heart tissues were viewed by confocal microscopy (NIKON, Eclipse C1). Rate of apoptosis was displayed as percentage of TUNEL positive nuclei in DAPI-stained nuclei.

### Western blot analysis
The left ventricular tissue was homogenized and digested in RIPA buffer (NCM, WB2100) with 1 mM PMSF (Byotime, ST507) and

phosphatase inhibitors (NCM, P002). The protein concentration was determined by BCA protein assay kit (NCM, WB6501). Western blot analysis was performed following a previously described method[77]. Each sample (30 ug of protein per lane) was separated by SDS–polyacrylamide gel (4–20%) (Beyotime, P0469S) electrophoresis, and the separated proteins were electrophoretically transferred from gel to a polyvinylidene fluoride membrane (Millipore, ISEQ00010). Afterward, the membranes were blocked in 5% skimmed milk in PBS with 0.1% Tween 20 (PBST) for 1 h at room temperature, followed by incubation with primary antibodies (Supplementary Table S1) at 4 °C overnight. The membrane was then incubated with peroxidase affini-Pure goat anti-rabbit IgG (H + L) (1:10000, Jackson, 111-035-144) and Peroxidase AffiniPure Goat Anti-Mouse IgG (H + L) (1:10000, Jackson, 115-035-146) at room temperature for 1 h. Finally, enhanced chemilu-minescence (ECL) reagent (MCE, HY-K1005) was used to detect the bands. GE IMAGEQUANT^MT 800 and ImageJ (version1.53a) was used for capturing and analyzing Western blot bands.

### RT-qPCR analysis
Total RNA was extracted from left ventricular tissues and HL-1 cell using the E.N.Z.A Total RNA Kit (Transgene, ER501-01-V2) and cDNA was synthesized using First-Strand cDNA Reverse Transcription SuperMix (Transgene, AU341-02-V2). Then, qPCR was performed with SYBR Green qPCR SuperMix (GeneCopoeia, QP001) on the ABI QuantStudio seven instrument. Primer sequences are listed in Sup-plementary Table S2. All qPCR results were calculated using the $2^{-\Delta\Delta CT}$ method.

### Plasma and myocardium noradrenaline, epinephrine and corti-costerone determination
Blood samples were collected from eye veins of mice. whole blood was placed in anticoagulant-treated tubes containing 1.6 mg/ml EDTA to obtain plasma. LV myocardial tissue (50 mg) was homo-genized using a homogenizer to produce a 10% tissue homogenate. NE, epinephrine (EPI) and cortisol (CORT) levels in plasma and tissue were measured by an enzyme-linked immunosorbent assay kit (Wuhan Huamei Biological Engineering, CSB-E07870m, CSB-E08679m, MLBIO, ML03158) following manufacturer's instructions. The OD value at 450 nm was detected using a microplate reader (BioTek Instruments, MQX 200).

### GSH/GSSG measurement
After myocardial tissues were weighed, the GSH/GSSG ratio was determined using the Glutathione Reductase/5,5'-Dithiobis-(2-nitro-benzoic acid) (DTNB) Cycling Assay Kit (Beyotime, S0055) according to the method recommended by the manufacturer and as previously reported[85]. The absorbance of the standards and samples was mea-sured at 412 nm. The absorbance value was fit to a standard curve of total glutathione and GSSG concentrations, after which the glu-tathione and GSSG concentrations in each sample were determined. The concentrations were converted to nmol/mg protein.

### Statistical analysis and reproducibility
All experiments were performed in at least three biological replicates, and each biological replicate contained three technical replicates. Data are presented as means ± S.E.M from at least three independent experiments. GraphPad Prism (Version 9.2.0) Statistics software was unitized for conducting statistical analysis. Statistical significance between two groups was determined by a two-tailed unpaired $t$ test. For multiple groups with equal variances, statistical significance of differences was evaluated using One-way ANOVA followed by Tukey's multiple comparisons tests or two-way ANOVA analysis followed by Bonferroni post hoc test. For multiple groups with unequal variances, Brown–Forsythe and Welch ANOVA tests followed by Dunnett's tests were adopted. $P$ values < 0.05 were considered statistically significant.

### Reporting summary
Further information on research design is available in the Nature Portfolio Reporting Summary linked to this article.

## Data availability
The authors declare that the data supporting the findings of this study are available within the paper and the supplementary information files. Source data are provided with this paper. Any additional raw data will be available from the corresponding author upon reasonable request. Source data are provided with this paper.

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

## Acknowledgements

This work was supported by the National Natural Science Foundation of China (No.82170291, No.81800058) E Wang and Lu Wang and National key research and development project of China (2020YFC2005300) to E Wang. Many thanks to Yongping Bai from Xiangya Hospital of Central South University for critically discussing the manuscript. The mouse image and the elements in Fig. 8 was provided by Figdraw (https://www.figdraw.com/#/.FigdrawexportID:TSUUI33276).

## Author contributions

N.C., L.G., L.W. and E.W. conceptualized the study. N.C., X.Z. and S.D. conducted experiments during the study and interpreted the results. L.W. analyzed data. E.W. supervised the study. N.C. and L.G. wrote the manuscript. L.W. contributed to the writing of the manuscript. E.W. reviewed and revised the manuscript.

## Competing interests

The authors declare no competing interests.
