## [Peer Review File · Nature Communications]

Reviewer Comments for Article: NCOMMS-23-55586

Summary Statement: Chen and colleagues examined the impact sleep fragmentation on myocardial injury in the male rat model and examined the role of sympathetic nerve activity mediated cardiomyocyte copper overload as a mechanism in cardiomyocyte apoptosis. The experimental design is robust and sleep fragmentation paradigm is adequate. Overall, sleep fragmentation worsened myocardial ischemia-reperfusion injury as a result of intracellular copper overload. The authors provide evidence that this may be mediated by sympathetic overactivity from sleep fragmentation. This is a very well-written and comprehensive manuscript with implications for sleep mediated cardiovascular disease. Minor methodological interpretative concerns are present in manuscript. Please see major and minor comments below to improve the present manuscript.

MAJOR COMMENTS

Methods

1. Please appropriately justify the exclusion of female mice. This is relevant because cardiovascular burden associated with short sleep is often of higher prevalence or worse within females compared to males. Perhaps the effect shown in this manuscript may be amplified in female mice.

Results/Discussion

1. HRV associated data is overinterpreted. The LF component of HRV is notoriously unreliable and often exhibits divergent changes compared to gold-standard direct sympathetic measurement. I would instead speak to the presumed reduction in HF HRV rather than the LF/HF ratio even though in this case it supports your sympathetic conclusions.
2. Sympathetic hyperreactivity appears to be mediated by an increase in synaptic nerve terminals, and while NE was increased in cardiac tissue, plasma NE was not increased. Please further elaborate on how sleep fragmentation triggers this neural remodeling to occur if known.
3. Please comment on whether increased synaptic nerve terminals with sleep fragmentation would be accompanied by increased sympathetic nerve activity. In response to TSD (human model), males often decrease SNA associated with a TSD-mediated hypertensive response. However, females increase SNA associated with increased BP from TSD. Could this phenomenon be at work in the male mice plasma NE was not increased after sleep fragmentation. Please elaborate in the discussion.

Reviewer #2 (Remarks to the Author):

The study by Chen and coworkers describes that sleep fragmentation aggravates ischemia/reperfusion injury in mice in vivo; the authors suggest that sleep fragmentation enhances norepinephrine concentrations thereby decreasing cardiomyocyte copper export (reduced sarcolemmal ATP7A expression), thereby increasing cardiomyocyte copper content leading to DLAT aggregation and induction of cuproptosis. Sympathetic denervation and copper chelator abolished the detrimental effect of sleep fragmentation. The data are of interest.

Major:

Overall the authors quite often use very small n-numbers (n=3 independent experiments), but perform multiple testings. With 3 comparisons, however, the number of independent experiments should exceed (n-1) meaning n should be at least 5. Even more, in some instances multiple t-tests should be replaced by ANOVA.

Figure 1: The authors should provide for copper concentrations in sham and I/R in control mice and mice with sleep fragmentation. Especially data following I/R are essential for their hypothesis.

Apoptosis was quantified by TUNEL staining. Here, the authors should state whether or not apoptosis was present only in infarcted areas or also in the border and control zones.

Even more, it would be helpful if authors provide tracing of necrotic tissue in figure 1G. What tissue was identified as being necrotic?

Figure 2: FDX1 is thought to be involved in lipoylation of DLAT (increase in FDX1 causes increased lipoylation). Lipoylated DLAT through copper binding at the lipoic acid sites then forms aggregates.

Here FDX1 (2H) was reduced by sleep fragmentation causing a decrease in LipDLAT (2F) but DLAT oligomers were increased (2D), interestingly only following I/R. Data on copper content would be central for understanding.

Figure 3: Sympathetic activity was increased by sleep fragmentation (HR increase) and norepinephrine content increased as well. How was heart rate in the I/R experiments (Fig 1) and how does increased heart rate contribute to I/R injury?

Increased norepinephrine in the presence of copper overload might undergo increased oxidation (through altered GSH oxidation and conjugation) leading to increased cardiomyocyte injury. What happened to the GSH/GSSG system with sleep fragmentation?

Figure 4: High concentration of catecholamines can induce increased reactive oxygen species formation and aldehyde concentration through monoamine oxidases. The authors should quantify formation of reactive oxygen species/aldehydes or repeat experiments in the presence of MAO inhibitors.

Figure 5: data for cell viability should be added (like in figure 4).

Figure 6: why is the profound decrease in copper in SCGx animals not associated with any effect on FDX1 or DLAT oligomers?

Figure 7: Increase in FDX1 and LipDLAT in the presence of reduced copper are associated with reduced DLAT oligomers and reduced cardiomyocyte damage. However, copper should be measured following I/R as well.

Minor:

Sleep deprivation and sympathetic neural responses in humans appear to be sex-dependent (PMID: 31149842). Why did authors only use male mice?

More recent reviews should be cited (PMID: 37989446; 37150036)

Reviewer #3 (Remarks to the Author):

The study presented is very interesting. The design of the study is suitable, merging common clinical measurements and a deep molecule characterization of the underlying mechanisms in an appropriate preclinical model of Myocardial Infarction (MI).

Overall, healthy elderly people sleep less than young adults and have more fragmented and unstable sleep, with frequent awakenings. This fragmentation is aggravated when there are banal anxiety problems and is then even more important when there are pathological conditions. When the condition of sleep fragmentation (SF) is associated with myocardial infarction, the effects are important for health.

Data presented are soundness and can have an important influence on the field but also important application for the daily lives of individuals with sleep disturbances and patients at risk or with cardiovascular diseases.

The study clearly shows that copper overload and cuproptosis are a key link in sympathetic hyperactivation that follows SF and aggravation of MI/RI in the SF mouse model.

However, in my opinion, the study might be further improved, as follows:

Methods:

1)The region of the heart that has been used for metal measures and all the WB RT-PCR and immunofluorescence are not detailed. It would be easier for the reader if the authors had more clearly indicated which infarcted areas of the heart have been analyzed in the biomolecular analyses and in the immunofluorescence imaging (left ventricle, but were exactly?, more details about if the tissue is in the TTC staining area or the close area)

Results

1)Figure 1: The schematic of the experimental protocol used can be improved with more details, reporting when Tissue inspections (metal measurements, WB, RT-PCR Immunofluorescence) have been done. This could be done graphically or in the figure legend indicating in the visit (D0, D1, D3, D7) the analyses done. Figure 1 which explains the experimental procedures does not show clearly the timeline. A clear image indicating the time with “day 0, day 1, day 3...”, or T0, T1, T3 instead of D would be better for the reader. Moreover, the images and diagrams of the results of Figure 1 do not refer to any D, also in the text it is not clearly indicated. This is the same for all the figures, in which it is not reported the time (day) of observation (or measurements). From the text, it seems that the day of the most represented Results is day 3.

2)Figure 3. Copper increases in the heart that undergoes MI after SF. It is not clearly indicated the time of the measures, how long after LAD? It is important to know if it is within 1 hour (hr) after LAD or after 3 days as suggested by the text. In fact, the Chevion's et al study (ref 50), quoted by the authors (PMID: 8430081), evidences in the first 35 min - 1 h after MI a substantial mobilization of copper in the coronary flow, with an increase of copper that is 8- to 9-fold higher than the pre-ischemic value. But afterward, the picture seems to change. At day 3 after LAD as the current study refers (but it is not clear) other authors, investigating only MI and not SF in MI, have reached completely different results. This is the case for a series of studies from James Kang's group. James Kang's group outlined the dynamics of Cu dysfunction by ligation of the left anterior descending coronary artery (LAD). James Kang's team demonstrated an increased release of myocardial Cu into the bloodstream, starting 1 day after LAD [PMID: 33653183, PMID: 32148090]. In other words, it has been observed a massive copper depletion from the heart after LAD that reaches 80% of heart copper in 10 days. Since the current authors do not show the condition without MI (a normal heart) the two studies are not strictly comparable, but at a glance, the dynamic of copper after MI seems not similar between the two studies. Furthermore, the copper release from the heart caused by MI seems due to the activity of the Cu transporter COMMD1 (MURR domain 1) [PMID: 33653183, PMID: 32148090], not considered in the current study. In another study by Ying Xiao Chen Li and colleagues [PMID: 33653183], the authors reconstructed the different phases of the release of Cu from the ischemic heart into the blood. Rats underwent LAD and serum Cu and ceruloplasmin levels were measured along with ATPase7B in the liver at days 1, 4, or 7 after surgery. Serum Cu concentrations increased significantly on day 4 after LAD ligation, accompanied by increases in serum ceruloplasmin concentration and activity. At the same time, Ceruloplasmin and ATPase7B levels in the liver increased significantly. The studies by James Kang et al show how starting from the 1st day after LAD there is an important release of copper from the heart, which feeds the serum copper reservoir (probably in the form of a copper not bound to ceruloplasmin) that then disappears in the liver and reappears in circulation as ceruloplasmin starting from day 4 after MI. overall this is in line with the

increase of serum copper and ceruloplasmin which is observed in large population studies on humans regarding MI and heart failure. On this basis, the biological mechanisms governing copper transport in the very early stages (35 min, 1 hr, 6, 12, and 24 hrs) of LAD seem pivotal. The authors could comment on the diversity of the two preclinical models on Copper dynamics after LAD and if published data fits with their model. Also, comment on the involvement of COMMD1 in the later stages of the MI process.

3)The authors postulated some hypotheses, which they correctly verified with their experiments. For example, on p. 7 line: 9 "...We hypothesized that inhibition of VPS35 expression was the key cause of the impaired intracellular copper transport induced by NE treatment...". Their results demonstrated VPS35's involvement in copper transport, but they did not consider that alternative processes may be involved, for example, anoxia (that triggers gene expression), or change in energy mechanisms from (oxidative phosphorylation to glycolysis and ATP deprivation), with possible effects on ATP7A functions. Authors can reframe the discussion by discussing these aspects

4) Superior cervical ganglionectomy (SCGx) rescued myocardial copper overload and exacerbated MI/RI in mice with SF (Figure 7). Why not use adrenergic receptor antagonists to block NE action? The use of beta-adrenergic receptor antagonists is common in cardiology. This has also a direct link with Beta-blockers drugs such as Bisoprolol which is specifically used in heart failure. In this way, the preclinical model of SF in MI would have been more adherent to the human clinical condition.

5) An inaccuracy of the study lies in the graphical representation in Figure 8 of ATP7A which is depicted as a receptor-like membrane protein, while ATP7A is an ion pump, which uses ATP to pump Cu^{2+} ions against the gradient, to the interior of the vesicles. The authors could modify the Figure appropriately.

6)It is likely that the anoxia caused by LAD has important effects on the production of ATP necessary for ATP7A to proceed with pumping copper into the vesicles. Still linked to anoxia the authors could comment on the effects of HIF which has been shown to interact with COMMD1(PMID: 17371845). These aspects could be commented on in the discussion.

7)In Figure 8, it should be clarified at least in Figure 8 Legend that the adrenergic receptors are G protein-coupled receptors (more specifically Adenylyl-cyclase, Phospholipase-C) with many target pathways so there is not a direct link between NE release and the inhibition of VPS35 and cargo vesicle transport in cardiomyocytes, leading to ATP7A-associated impairment of copper transport.

Discussion

1)As already mentioned, it is appropriate for the authors to take into consideration and comment on the studies of James Kang's group from Sichuan. Furthermore, they should mention and integrate in their discussion the existence of more complex processes that are certainly involved in MI, such as Hypoxia and HIF activation.

2) for example, the authors state "...In conclusion, the inhibition of VPS35 expression by NE treatment caused copper overload in HL-1 cells mainly through disruption of fusion between copper-transporting vesicles and the plasma membrane..." This conclusion is too simple, since some processes in the middle should mentioned, and others may be envisaged (hypoxia, and gene expression linked to hypoxia).

Reviewer #1:

Chen and colleagues examined the impact sleep fragmentation on myocardial injury in the male rat model and examined the role of sympathetic nerve activity mediated cardiomyocyte copper overload as a mechanism in cardiomyocyte apoptosis. The experimental design is robust and sleep fragmentation paradigm is adequate. Overall, sleep fragmentation worsened myocardial ischemia-reperfusion injury as a result of intracellular copper overload. The authors provide evidence that this may be mediated by sympathetic overactivity from sleep fragmentation. This is a very well-written and comprehensive manuscript with implications for sleep mediated cardiovascular disease. Minor methodological interpretative concerns are present in manuscript. Please see major and minor comments below to improve the present manuscript.

MAJOR COMMENTS

Methods

1. Please appropriately justify the exclusion of female mice. This is relevant because cardiovascular burden associated with short sleep is often of higher prevalence or worse within females compared to males. Perhaps the effect shown in this manuscript may be amplified in female mice.

Our response:

We appreciate the reviewer for raising this important question. Indeed, we considered the influence of mouse sex before the start of our study. Notably, female mice undergo cyclical fluctuations in estrogen levels. Studies have indicated that estrogen levels are correlated with diminished fibrosis, decreased oxidative stress, enhanced mitochondrial function, mitigated cardiac hypertrophy, and augmented angiogenesis following cardiac injury in female mice^{1, 2, 3, 4, 5}. Therefore, the use of female mice could affect the consistency of the intragroup results and even interfere with the study outcomes. Besides, our study primarily focused on investigating how the sympathetic nervous system regulates copper metabolism in cardiomyocytes. Consequently, we ultimately

chose male mice for our research.

We sincerely apologize for missing such crucial information in the abstract and have added a clarification about the sex of mice used to the abstract (page 1, line 7). In the discussion section of our manuscript, we included information about the potential impact of the sex of the mice used on our study and the associated research limitations (page 18, line 389 to 395).

Results/Discussion

1. HRV associated data is overinterpreted. The LF component of HRV is notoriously unreliable and often exhibits divergent changes compared to goldstandard direct sympathetic measurement. I would instead speak to the presumed reduction in HF HRV rather than the LF/HF ratio even though in this case it supports your sympathetic conclusions.

Our response:

We are very grateful for your valuable suggestion. We replaced the LH/HF ratio with the normalized HF (HFnorm) according to the previous research^{6,7}. Consistent with the earlier findings, the significantly diminished HFnorm in SF mice indicates a substantial increase in sympathetic nervous system activity associated with sleep fragmentation. Relevant changes were made to the methods (page 24, line 511 to 513,) and results sections (page 6, line 113 to 114, Fig. 3J).

2. Sympathetic hyperreactivity appears to be mediated by an increase in synaptic nerve terminals, and while NE was increased in cardiac tissue, plasma NE was not increased. Please further elaborate on how sleep fragmentation triggers this neural remodeling to occur if known.

Our response:

We appreciate the reviewer's important question. After an extensive review of the literature, we found that the neural remodeling promoted by fragmented sleep may involve mechanisms such as neural plasticity and the stimulation of neurotrophic factors. Neural plasticity is the inherent ability of the nervous system to adapt to

external and internal signals^{8,9}. Our experimental data revealed enhanced signaling from the SCG. On the one hand, there was an increase in sympathetic neurotransmitter synthesis in the SCG of SF mice (Fig. S3I-S3K). On the other hand, complementary experimental data suggest that neurons in the SCG are hyperexcitable during fragmented sleep (Fig. S3L-S3M), which may be due to stimulation by upstream signals. Therefore, long-term fragmented sleep could enhance neural conduction, synaptic transmission, and neurotransmitter release in the PVN-SCG-cardiac sympathetic pathway^{10,11}, which represents the most likely mechanism for the activations of cardiac sympathetic remodeling. Moreover, neurotrophic factors such as neurotrophic factor 3 (NTF3) play crucial roles in neural remodeling^{12, 13}. Our results revealed increased NTF3 expression in the SCG of SF mice (Fig. S3H), suggesting it might be another potential mechanism for sympathetic remodeling.

3. Please comment on whether increased synaptic nerve terminals with sleep fragmentation would be accompanied by increased sympathetic nerve activity. In response to TSD (human model), males often decrease SNA associated with a TSD-mediated hypertensive response. However, females increase SNA associated with increased BP from TSD. Could this phenomenon be at work in the male mice plasma NE was not increased after sleep fragmentation. Please elaborate in the discussion.

Our response:

We appreciate your advice. Our study demonstrated that sleep fragmentation indeed leads to increased sympathetic nerve activity in male mice. Activation of the SCG (Fig. S3I-S3M), increased heart rate and cardiac sympathetic innervation (Fig. 3H-3M), and elevated cardiac norepinephrine (NE) and plasma epinephrine (EPI) levels (Fig. 3N, Fig. S3D) all indicated prolonged fragmented sleep as a trigger for sympathetic overactivity.

Compared with TSD models in studies you have mentioned^{14, 15}, we found that human TSD established by 24 hours of total sleep deprivation, whereas our model involves 16 weeks of fragmented sleep. Thus, these two sleep disorder models represent acute sleep deprivation and chronic sleep disturbances, respectively. It is reasonable to expect that

they involve distinct underlying physiological mechanisms. Moreover, our study could align more with observations of male workers who experience more than 5 years of rotating shift work or male rats subjected to 8 weeks of sleep deprivation, which both exhibited sympathetic overactivity^{16, 17}.

We have included relevant information in the discussion section (page 10, line 217 to 221).

Reviewer #2:

The study by Chen and coworkers describes that sleep fragmentation aggravates ischemia/reperfusion injury in mice in vivo; the authors suggest that sleep fragmentation enhances norepinephrine concentrations thereby decreasing cardiomyocyte copper export (reduced sarcolemmal ATP7A expression), thereby increasing cardiomyocyte copper content leading to DLAT aggregation and induction of cuproptosis. Sympathetic denervation and copper chelator abolished the detrimental effect of sleep fragmentation. The data are of interest.

Major:

1. Overall, the authors quite often use very small n-numbers (n=3 independent experiments) but perform multiple testings. With 3 comparisons, however, the number of independent experiments should exceed (n-1) meaning n should be at least 5. Even more, in some instances multiple t-tests should be replaced by ANOVA.

Our response:

We greatly appreciate your valuable suggestion, which has significantly improved the robustness and reliability of our study. We have conducted additional experiments to increase the number of independent experiments, increased the sample sizes to 5 or more, reconducted the statistical analyses, remade the graphs, and used ANOVA for the statistical analyses according to your advice. Additional data have been added to Figs. 1, 2, 4, 6, and 7 and to the supplementary material.

2. Figure 1: The authors should provide for copper concentrations in sham and I/R in control mice and mice with sleep fragmentation. Especially data following I/R are essential for their hypothesis.

Our response:

We are very grateful for your valuable suggestion. We added data regarding changes in copper content after MI/RI. The results demonstrated a significant increase in myocardial copper content following MI/RI, with even greater increase in SF mice after MI/RI. These findings are consistent with a previous study that explored elevated copper levels after brain I/R¹⁸. This finding suggests that copper overload in SF mice occurs before and after MI/RI and may indeed play a crucial role in subsequent injury. We have included these data in Figure 3A, on page 5, line 92.

3. Figure 1: Apoptosis was quantified by TUNEL staining. Here, the authors should state whether or not apoptosis was present only in infarcted areas or also in the border and control zones. Even more, it would be helpful if authors provide tracing of necrotic tissue in figure 1G. What tissue was identified as being necrotic?

Our response:

We thank the reviewer for the kind suggestion. We observed that cell apoptosis primarily occurs in the infarcted area, with some occurring in the marginal area, which is consistent with the findings reported in a previous study¹⁹. We have added an explanation to the legend of Figure 1I and included this information in the Results section (page 4, line 73).

Figure 1G displays the results of myocardial slices stained with TTC (2,3,5-triphenyltetrazolium chloride). TTC staining is a commonly used method to assess tissue viability and metabolic activity^{20, 21}. The white areas indicate regions where cellular metabolism is reduced or incapable of reducing TTC due to factors such as cell death, ischemia, or hypoxia. Therefore, the white areas often indicate necrotic tissue in the infarcted region²². We have added this explanation to the legend of Fig. 1G.

4. Figure 2: FDX1 is thought to be involved in lipoylation of DLAT (increase in FDX1 causes increased lipoylation). Lipoylated DLAT through copper binding at the lipoic acid sites then forms aggregates. Here FDX1 (2H) was reduced by sleep fragmentation causing a decrease in LipDLAT (2F) but DLAT oligomers were increased (2D), interestingly only following I/R. Data on copper content would be central for understanding.

Our response:

Thank you very much for your valuable suggestion. In the study conducted by Peter Tsvetkov and colleagues²³, increasing copper concentrations dose-dependently inhibited the expression of FDX1 and Lip-DALT, accompanied by an increase in DALT oligomers, leading to enhanced cuproptosis. This indicates that high copper levels not only suppress the expression of FDX1 and Lip-DLAT but also result in an increase in DLAT oligomers. In our study, there was a significant increase in copper levels after MI/RI (added to Figure 3A). High copper levels indeed inhibit the expression of FDX1 and Lip-DLAT, which is consistent with the findings of Peter Tsvetkov and colleagues²³. As you correctly pointed out, determining the data on copper content after MI/RI is crucial for comprehending this process, and the results have been included in Figure 3A, on page 5, line 92.

5. Figure 3: Sympathetic activity was increased by sleep fragmentation (HR increase) and norepinephrine content increased as well. How was heart rate in the I/R experiments (Fig 1) and how does increase heart rate contribute to I/R injury?

Our response:

Thank you very much for your important suggestion. Despite our previous oversight regarding HR after I/R, we have added relevant information to our study. Cardiac electrophysiology telemetry data showed that the HR was significantly increased 3 days after MI/RI, regardless of whether the mice were in the control group or the SF group. Additionally, the SF group still exhibited a higher HR compared to the control group after MI/RI. This finding is consistent with the results reported by Belem Yoval-Sánchez, who reported an increase in HR after MI/RI in rats²⁴. This information has

been added to Fig. S3C, on page 6, line 112 to 115.

Both baseline HR elevation and increased HR after I/R can exacerbate MI/RI^{25, 26}. A high HR could lead to increased myocardial oxygen demand, vigorous energy metabolism, and depletion of ATP and phosphocreatine reserves²⁷. Additionally, it can reduce coronary artery perfusion by shortening diastole²⁸. Therefore, increased HR may contribute to an imbalance of myocardial oxygen supply and demand, thereby exacerbating MI/RI.

6. Increased norepinephrine in the presence of copper overload might undergo increased oxidation (through altered GSH oxidation and conjugation) leading to increased cardiomyocyte injury. What happened to the GSH/GSSG system with sleep fragmentation?

Our response:

Thank you for your valuable suggestion. As you mentioned, we have added an assessment of the mouse GSH/GSSG system to our study (Fig. S4A-S4C). We detected elevated GSSG levels and a decreased GSH/GSSG ratio in the myocardium of SF mice. This indicates that the myocardium of SF mice exhibits increased oxidative changes. Numerous studies have also demonstrated similar changes in the hippocampus of sleep-deprived rats^{29, 30}. The results have been included in Fig. S4A-S4C, on page 7, line 135 to 136.

7. Figure 4: High concentration of catecholamines can induce increased reactive oxygen species formation and aldehyde concentration through monoamine oxidases. The authors should quantify formation of reactive oxygen species/aldehydes or repeat experiments in the presence of MAO inhibitors.

Our response:

We are very grateful for your valuable suggestion, and we have repeated the experiments in the presence of an MAO inhibitor (Fig. S4D-S4O). The results showed that the increased cellular copper and cuproptosis brought about by NE was not attenuated in the presence of the MAO inhibitor. The above results indicate that MAOs

are not an intermediate in the process by which NE leads to copper overload and increased cuproptosis in cardiomyocytes. The results have been included in Figure S4D-S4O, on page 7, line 133 to 141.

8. *Figure 5: data for cell viability should be added (like in figure 4).*

Our response:

We appreciate for your suggestion and have added the results of the CCK-8 assay for vps35-overexpressing cells and control cells treated with different concentrations of NE to Fig. 5O.

9. *Figure 6: why is the profound decrease in copper in SCGx animals not associated with any effect on FDX1 or DLAT oligomers?*

Our response:

Thank you for your question. Importantly, our experimental data indicate that the reduction in copper levels induced by SCGx indeed affects the expression of FDX1 and DLAT oligomers.

Regarding FDX1, our data, along with those of Peter Tsvetkov, suggest that changes in copper levels may feedback regulate the expression of FDX1. Specifically, chronic copper overload caused by SF inhibits the expression of FDX1, while SCGx, copper chelators, can slow down the feedback downregulation of FDX1 by lowering copper levels.

The findings regarding the changes in DLAT oligomers were contradictory. SCGx in SF mice significantly alleviated the increase in DLAT oligomers after MI/RI, whereas the same effect was not observed in mice with normal sleep. However, we believe that this difference was mainly due to the 16 weeks of chronic copper accumulation in SF mice. Therefore, the increase in DLAT oligomers in the SF+IR group was a comprehensive effect of acute and chronic copper accumulation, with a higher baseline copper load than that in the Sham+IR group. The experimental data also indicate that prolonged chronic copper accumulation may increase the expression of copper transport proteins, including ATP7A, in myocardial cells. When SCGx normalizes

copper efflux in myocardial cells, the efficiency of copper ion efflux should also be higher in SF mice. Therefore, SCGx-induced alleviation in cuproptosis is more pronounced in the SF+IR mouse than in the control group, whether from the perspective of baseline copper load or improved efficiency of copper ion efflux. Additionally, to ensure the consistency of the results detection time, copper ion detection, 3 days after SCGx, shows a continuous efflux of intracellular copper to lower levels in both groups. However, the peak difference in the formation of DLAT oligomers induced by MI/RI may have occurred at an earlier time point, which may also lead to inconsistency in the trends of copper ion levels and DLAT oligomer levels.

We have included relevant information in the discussion section (page 13, line 271 to 293).

10. Figure 7: Increase in FDX1 and LipDLAT in the presence of reduced copper are associated with reduced DLAT oligomers and reduced cardiomyocyte damage. However, copper should be measured following I/R as well.

Our response:

Thank you for your valuable suggestion. We have included additional experimental data in this section and found that TTM alleviated the increase in copper levels after MI/RI, which is consistent with our previous findings. The results are included in Figure 7B.

Minor:

1. Sleep deprivation and sympathetic neural responses in humans appear to be sex dependent (PMID: 31149842). Why did authors only use male mice?

Our response:

We appreciate the reviewer for raising this important question. We considered the influence of mouse sex before the start of our study. Female mice experience cyclical fluctuations in estrogen levels, and estrogen has been shown to be associated with reduced fibrosis, decreased oxidative stress, improved mitochondrial function, attenuated cardiac hypertrophy, and increased angiogenesis after cardiac injury^{1, 2, 3, 4, 5}. Besides, our study primarily focused on investigating how the sympathetic nervous

system regulates copper metabolism. Consequently, we ultimately chose male mice for our research. We included information about the potential impact of the sex of the mice we used and the associated research limitations (page 18, line 389 to 395).

2. More recent reviews should be cited (PMID: 37989446; 37150036)

Our response:

Thank you very much for your suggestions. We have carefully reviewed the literature you provided and have added them to the reference list as references 7 and 58, respectively.

Reviewer #3:

The study presented is very interesting. The design of the study is suitable, merging common clinical measurements and a deep molecule characterization of the underlying mechanisms in an appropriate preclinical model of Myocardial Infarction (MI).

Overall, healthy elderly people sleep less than young adults and have more fragmented and unstable sleep, with frequent awakenings. This fragmentation is aggravated when there are banal anxiety problems and is then even more important when there are pathological conditions. When the condition of sleep fragmentation (SF) is associated with myocardial infarction, the effects are important for health. Data presented are soundness and can have an important influence on the field but also important application for the daily lives of individuals with sleep disturbances and patients at risk or with cardiovascular diseases. The study clearly shows that copper overload and cuproptosis are a key link in sympathetic hyperactivation that follows SF and aggravation of MI/RI in the SF mouse model. However, in my opinion, the study might be further improved, as follows:

Methods:

1) The region of the heart that has been used for metal measures and all the WB RT-PCR and immunofluorescence are not detailed. It would be easier for the reader if the authors had more clearly indicated which infarcted areas of the heart have been analyzed in the biomolecular analyses and in the immunofluorescence imaging (left ventricle, but were exactly? more details about if the tissue is in the TTC staining area or the close area)

Our response:

We sincerely apologize sorry for omitting such crucial information. We have added details to the methods and figure legends based on your suggestions.

For the metal level measurements, samples were of the entire left ventricle below the ligature suture plane. This additional detail has been included in the subsection on metal measurement in the methods section (page 23, line 492 to 493).

For RT-PCR and Western blot analyses, samples were mainly taken from the left ventricle below the ligature, including the ischemic area (TTC-stained area) and a small portion of the marginal tissue. This information has been added to the subsection on tissue preparation in the methods section (page 25 to 26, line 548 to 554).

TUNEL-positive regions were scattered within the TTC-stained area and its vicinity. The DLAT oligomers-positive regions, as evident from the gross image, were predominantly concentrated in the middle layer of the left ventricle, centered on the TTC-stained region. Images of ATP7A and WGA co-staining in the anterolateral wall of the mid-lower segment of the left ventricle were captured. We have explicitly specified these locations in the figure legends (Figs. 1I, 2A, 3G).

Results

1) Figure 1: The schematic of the experimental protocol used can be improved with more details, reporting when Tissue inspections (metal measurements, WB, RT-PCR Immunofluorescence) have been done. This could be done graphically or in the figure legend indicating in the visit (D0, D1, D3, D7) the analyses done. Figure 1 which explains the experimental procedures does not show clearly the timeline. A clear image indicating the time with “day 0, day 1, day 3...”, or T0, T1, T3 instead

of D would be better for the reader. Moreover, the images and diagrams of the results of Figure 1 do not refer to any D, also in the text it is not clearly indicated. This is the same for all the figures, in which it is not reported the time (day) of observation (or measurements). From the text, it seems that the day of the most represented Results is day 3.

Our response:

Thank you for your valuable feedback and suggestion. We have made the suggested changes to all the schematic diagrams of the experimental protocol. This includes using "Day0, Day3, and Day7" to indicate the observation time points, instead of "D0, D3, and D7." Additionally, we have provided explanations for the various methods used for tissue analysis, such as metal measurements, WB, RT-qPCR and immunofluorescence (Fig. 1A, 6A, 7A).

Regarding the schematic diagram in Figure 1 that you mentioned, the observation time for all the charts in Figure 1 was indeed 3 days after MI/RI. We sincerely apologize for the oversight in our previous submission. We have rectified this error in the newly submitted figure. Once again, we appreciate your attention and important reminder.

2) Figure 3. Copper increases in the heart that undergoes MI after SF. It is not clearly indicated the time of the measures, how long after LAD? It is important to know if it is within 1 hour (hr) after LAD or after 3 days as suggested by the text. In fact, the Chevion's et al study (ref 50), quoted by the authors (PMID: 8430081), evidences in the first 35 min - 1 h after MI a substantial mobilization of copper in the coronary flow, with an increase of copper that is 8- to 9-fold higher than the pre-ischemic value. But afterward, the picture seems to change. At day 3 after LAD as the current study refers (but it is not clear) other authors, investigating only MI and not SF in MI, have reached completely different results. This is the case for a series of studies from James Kang's group. James Kang's group outlined the dynamics of Cu dysfunction by ligation of the left anterior descending coronary artery (LAD). James Kang's team demonstrated an increased release of myocardial Cu into the bloodstream, starting 1 day after LAD [PMID: 33653183, PMID: 32148090]. In other words, it has been

observed a massive copper depletion from the heart after LAD that reaches 80% of heart copper in 10 days. Since the current authors do not show the condition without MI (a normal heart) the two studies are not strictly comparable, but at a glance, the dynamic of copper after MI seems not similar between the two studies. Furthermore, the copper release from the heart caused by MI seems due to the activity of the Cu transporter COMMD1 (MURR domain 1) [PMID: 33653183, PMID: 32148090], not considered in the current study. In another study by Ying Xiao Chen Li and colleagues [PMID: 33653183], the authors reconstructed the different phases of the release of Cu from the ischemic heart into the blood. Rats underwent LAD and serum Cu and ceruloplasmin levels were measured along with ATPase7B in the liver at days 1, 4, or 7 after surgery. Serum Cu concentrations increased significantly on day 4 after LAD ligation, accompanied by increases in serum ceruloplasmin concentration and activity. At the same time, Ceruloplasmin and ATPase7B levels in the liver increased significantly. The studies by James Kang et al show how starting from the 1st day after LAD there is an important release of copper from the heart, which feeds the serum copper reservoir (probably in the form of a copper not bound to ceruloplasmin) that then disappears in the liver and reappears in circulation as ceruloplasmin starting from day 4 after MI. overall this is in line with the increase of serum copper and ceruloplasmin which is observed in large population studies on humans regarding MI and heart failure. On this basis, the biological mechanisms governing copper transport in the very early stages (35 min, 1 hr, 6, 12, and 24 hrs) of LAD seem pivotal. The authors could comment on the diversity of the two preclinical models on Copper dynamics after LAD and if published data fits with their model. Also, comment on the involvement of COMMD1 in the later stages of the MI process.

Our response:

Thank you very much for your valuable questions and friendly suggestions. Considering the discrepancy between the dynamic changes in myocardial copper content after injury in the LAD artery ligation and MI model studies and the results of our research, we would like to emphasize that the LAD artery ligation model is a model

of ischemic heart disease involving complete obstruction of coronary blood flow, while the cardiac injury model we used throughout the study is a myocardial ischemia-reperfusion injury (MI/RI) model. In other words, in our model, blood flow was restored to the heart 30 minutes after ischemia, and we focused on observing the impact of reperfusion injury on cardiac function. The observation time point was the third day after restoration of blood perfusion to the heart. Therefore, our results may not be directly comparable to those of the study you mentioned.

Furthermore, a considerable body of research in MI/RI^{31, 32, 33, 34} suggests that increased copper load exacerbates MI/RI, and that interventions such as promoting copper excretion and copper chelation may have potential therapeutic value in MI/RI, which aligns with our research findings. Our previous experimental data, as well as the additional experimental data provided in this revision, consistently indicate that SF mice exhibit a copper-overload phenotype both before and after MI/RI, further supporting the idea that SF exacerbates MI/RI by increasing cardiac copper overload. Although there was no direct measurement of cardiac copper content after MI/RI before our study, a recent study on cerebral I/R injury also revealed a significant increase in copper content after I/R injury¹⁸.

We acknowledge the importance of the copper transport protein COMMD1 (MURR domain 1) in copper transport, and we sincerely apologize for overlooking this protein in our experiments screening for potential intervention targets for SF-induced myocardial cell copper overload. In the revised manuscript, we have added experiments to measure the expression levels of COMMD1 in animal tissues, and the results indicate that SF does not affect the expression of COMMD1 in the heart (Fig. S5B).

Once again, we appreciate your valuable insights and suggestions, and we have addressed the issues you raised in the discussion section of the revised manuscript (page 15, line 326 to 328, and page 17, line 363 to 387).

3)The authors postulated some hypotheses, which they correctly verified with their experiments. For example, on p. 7 line: 9 “...We hypothesized that inhibition of VPS35 expression was the key cause of the impaired intracellular copper transport

induced by NE treatment...". Their results demonstrated VPS35's involvement in copper transport, but they did not consider that alternative processes may be involved, for example, anoxia (that triggers gene expression), or change in energy mechanisms from (oxidative phosphorylation to glycolysis and ATP deprivation), with possible effects on ATP7A functions. Authors can reframe the discussion by discussing these aspects.

Our response:

We appreciate your valuable advice, which was crucial for improving our manuscript. We apologize that we overlooked potentially important upstream molecular biology mechanisms in the discussion section. Indeed, the discussion section of the previous version of the manuscript focused on analyzing the potential mechanisms through which targeting VPS35 could alleviate copper overload, because our experimental data validated that the overexpression of VPS35 significantly mitigated the copper overload phenotype induced by sympathetic signals in myocardial cells (Fig. 5A-5O), which indicated that the copper transport function of ATP7A itself may not be significantly influenced by factors other than VPS35 in our model. However, the upstream mechanisms that mediate the inhibition of VPS35 expression caused by sleep disorders may involve the hypoxia as you mentioned, which still requires further in-depth discussion.

Through an extensive literature review, we found that while there is limited research on the regulation of VPS35 expression in cardiomyocytes, cutting-edge studies in the field of oncology have provided some potential insights. Some articles suggest that inhibiting the heat shock protein 90 (Hsp90) can promote the expression of VPS35 by upregulating Bclaf-1, thereby facilitating extracellular vesicle release in liver cancer cells³⁵. Since HSP90 upregulation was observed in response to cellular stress or hypoxia signals^{36, 37}, sleep disorders or I/R injury may mediate copper overload in myocardial cells through the Hsp90-Bclaf1-Vps35 pathway, providing a crucial research direction for our subsequent investigations into the underlying mechanisms involved. This information has been added to the discussion section of the manuscript (page 16 to 17, line 351 to 363).

4) Superior cervical ganglionectomy (SCGx) rescued myocardial copper overload and exacerbated MI/RI in mice with SF (Figure 7). Why not use adrenergic receptor antagonists to block NE action? The use of beta-adrenergic receptor antagonists is common in cardiology. This has also a direct link with Beta-blockers drugs such as Bisoprolol which is specifically used in heart failure. In this way, the preclinical model of SF in MI would have been more adherent to the human clinical condition.

Our response:

We are very grateful for your question. We ultimately chose to use SCGx rather than adrenergic receptor antagonists for two main reasons. First, we aimed to interrupt this structural connection and opted for the anatomical disconnection of SCG to precisely sever the signaling cascade upstream, while minimally affecting the systemic sympathetic impact.

Second, norepinephrine (NE) is a nonselective adrenergic receptor agonist. β -Adrenergic receptor blockers (such as bisoprolol, a β 1-adrenergic receptor blocker) typically selectively block β 1 receptors and may not completely inhibit the actions of NE. Moreover, systemic administration of such drugs may introduce systemic interference, such as by simultaneously suppressing the adrenal response in myocardial ischemia^{38, 39}. Additionally, there is a possibility of interference due to the drug concentration or the drug level in the blood following a single dose. Considering these factors, we chose the anatomical intervention method, which has a lesser systemic impact and is more precise and comprehensive in its effect, despite its greater technical difficulty.

5) An inaccuracy of the study lies in the graphical representation in Figure 8 of ATP7A which is depicted as a receptor-like membrane protein, while ATP7A is an ion pump, which uses ATP to pump Cu²⁺ ions against the gradient, to the interior of the vesicles. The authors could modify the Figure appropriately.

Our response:

We sincerely apologize for the inaccurate representation of the molecular schematic of

ATP7A and sincerely appreciate the questions you raised. We have modified to the graphical representation of ATP7A in Figure 8.

6)It is likely that the anoxia caused by LAD has important effects on the production of ATP necessary for ATP7A to proceed with pumping copper into the vesicles. Still linked to anoxia the authors could comment on the effects of HIF which has been shown to interact with COMMD1(PMID: 17371845). These aspects could be commented on in the discussion.

Our response:

We appreciate your thoughtful insights, and we sincerely thank you for your valuable suggestions. Indeed, while there are some distinctions in pathological mechanisms between the MI/RI model and the LAD artery ligation model, it is undeniable that, in addition to sympathetic hyperactivity, the molecular and biological changes induced by hypoxia in the MI/RI model triggered could be some of the crucial upstream mechanisms affecting the copper transport pathway. We acknowledge the omission of the potential impact of hypoxia on the copper transport pathway and the underlying mechanisms in our discussion. We apologize for this oversight and have added relevant information to the discussion section of the revised manuscript (page16 to 18, line 351 to 387).

7)In Figure 8, it should be clarified at least in Figure 8 Legend that the adrenergic receptors are G protein-coupled receptors (more specifically Adenylyl-cyclase, Phospholipase-C) with many target pathways so there is not a direct link between NE release and the inhibition of VPS35 and cargo vesicle transport in cardiomyocytes, leading to ATP7A-associated impairment of copper transport.

Our response:

We sincerely apologize for the inaccuracies in Figure 8, and we sincerely appreciate the questions you raised. In the new schematic, we have changed the solid arrow from NE to VPS35 to a dashed arrow. This modification signifies that this cellular phenotype is an observation made by us and does not imply a direct regulatory role of adrenergic

receptors on VPS35 or ATP7A-associated impairment of copper transport. The underlying molecular mechanisms involved in this regulatory relationship require further elucidation. Additionally, we have adhered to your suggestion and added corresponding explanatory notes to the legend.

Discussion

1) As already mentioned, it is appropriate for the authors to take into consideration and comment on the studies of James Kang's group from Sichuan. Furthermore, they should mention and integrate in their discussion the existence of more complex processes that are certainly involved in MI, such as Hypoxia and HIF activation.

Our response:

We greatly appreciate your valuable advice. As suggested in your previous questions 2, 3, and 6, we have included an additional paragraph in the discussion section. This paragraph compares the copper content after MI/RI observed in our study with the results of the James Kang group. Furthermore, comprehensively assessed the impact of various factors, such as hypoxia, HIF activation, and the upregulation of COMMD1 expression, on the early and long-term effects of myocardial injury. Please refer to line 351 to 387 for specific details.

2) for example, the authors state “...In conclusion, the inhibition of VPS35 expression by NE treatment caused copper overload in HL-1 cells mainly through disruption of fusion between copper-transporting vesicles and the plasma membrane...” This conclusion is too simple, since some processes in the middle should mentioned, and others may be envisaged (hypoxia, and gene expression linked to hypoxia).

Our response:

Thank you very much for your valuable suggestions for the manuscript. Regarding the conclusion, we previously adopted a more conservative approach by drawing conclusions based on existing experimental data. After carefully considering your advice, we have incorporated additional content in the discussion section, discussing

hypothetical intrinsic mechanisms (page 16 to 18, line 351 to 387).

1. Aryan L, *et al.* The Role of Estrogen Receptors in Cardiovascular Disease. *Int J Mol Sci* **21**, (2020).
2. Heger J, Szabados T, Brosinsky P, Bencsik P, Ferdinandy P, Schulz R. Sex Difference in Cardioprotection against Acute Myocardial Infarction in MAO-B Knockout Mice In Vivo. *Int J Mol Sci* **24**, (2023).
3. Iorga A, Cunningham CM, Moazeni S, Ruffenach G, Umar S, Eghbali M. The protective role of estrogen and estrogen receptors in cardiovascular disease and the controversial use of estrogen therapy. *Biol Sex Differ* **8**, 33 (2017).
4. Murphy E. Estrogen signaling and cardiovascular disease. *Circ Res* **109**, 687-696 (2011).
5. Yang XP, Reckelhoff JF. Estrogen, hormonal replacement therapy and cardiovascular disease. *Curr Opin Nephrol Hypertens* **20**, 133-138 (2011).
6. Khor KH, Moore TA, Shiels IA, Greer RM, Arumugam TV, Mills PC. A Potential Link between the C5a Receptor 1 and the beta1-Adrenoreceptor in the Mouse Heart. *PLoS One* **11**, e0146022 (2016).
7. Chen YS, Lin YY, Shih CC, Kuo CD. Relationship Between Heart Rate Variability and Pulse Rate Variability Measures in Patients After Coronary Artery Bypass Graft Surgery. *Front Cardiovasc Med* **8**, 749297 (2021).
8. Debanne D, Inglebert Y, Russier M. Plasticity of intrinsic neuronal excitability. *Curr Opin Neurobiol* **54**, 73-82 (2019).
9. Carey L, *et al.* Finding the Intersection of Neuroplasticity, Stroke Recovery, and Learning: Scope and Contributions to Stroke Rehabilitation. *Neural Plast* **2019**, 5232374 (2019).
10. Liu Z, *et al.* Role of ventrolateral part of ventromedial hypothalamus in post-myocardial infarction cardiac dysfunction induced by sympathetic nervous system. *J Mol Cell Cardiol* **184**, 37-47 (2023).
11. Wang Y, *et al.* Sympathetic Nervous System Mediates Cardiac Remodeling After Myocardial Infarction in a Circadian Disruption Model. *Front Cardiovasc Med* **8**, 668387 (2021).
12. Cuello AC. Experimental neurotrophic factor therapy leads to cortical synaptic remodeling and compensates for behavioral deficits. *J Psychiatry Neurosci* **22**, 46-55 (1997).
13. Chang YX, *et al.* Intramuscular Injection of Adenoassociated Virus Encoding Human Neurotrophic Factor 3 and Exercise Intervention Contribute to Reduce Spasms after Spinal Cord Injury. *Neural Plast* **2019**, 3017678 (2019).
14. Carter JR, Fonkoue IT, Greenlund IM, Schwartz CE, Mokhlesi B, Smoot CA. Sympathetic neural responsiveness to sleep deprivation in older adults: sex differences. *Am J Physiol Heart Circ Physiol* **317**, H315-H322 (2019).
15. Carter JR, Durocher JJ, Larson RA, DellaValla JP, Yang H. Sympathetic neural responses to 24-hour sleep deprivation in humans: sex differences. *Am J Physiol Heart Circ Physiol* **302**, H1991-1997 (2012).
16. Wehrens SM, Hampton SM, Skene DJ. Heart rate variability and endothelial function after sleep deprivation and recovery sleep among male shift and non-shift workers. *Scand J Work Environ Health* **38**, 171-181 (2012).

17. Wang Z, *et al.* Insomnia Promotes Hepatic Steatosis in Rats Possibly by Mediating Sympathetic Overactivation. *Front Physiol* **12**, 734009 (2021).
18. Guo Q, Ma M, Yu H, Han Y, Zhang D. Dexmedetomidine enables copper homeostasis in cerebral ischemia/reperfusion via ferredoxin 1. *Ann Med* **55**, 2209735 (2023).
19. Zhang Y, *et al.* The long noncoding RNA lncCIRBIL disrupts the nuclear translocation of Bclaf1 alleviating cardiac ischemia-reperfusion injury. *Nat Commun* **12**, 522 (2021).
20. Cole DJ, Drummond JC, Ghazal EA, Shapiro HM. A reversible component of cerebral injury as identified by the histochemical stain 2,3,5-triphenyltetrazolium chloride (TTC). *Acta Neuropathol* **80**, 152-155 (1990).
21. Isayama K, Pitts LH, Nishimura MC. Evaluation of 2,3,5-triphenyltetrazolium chloride staining to delineate rat brain infarcts. *Stroke* **22**, 1394-1398 (1991).
22. Lee TL, *et al.* Conditioned medium from adipose-derived stem cells attenuates ischemia/reperfusion-induced cardiac injury through the microRNA-221/222/PUMA/ETS-1 pathway. *Theranostics* **11**, 3131-3149 (2021).
23. Tsvetkov P, *et al.* Copper induces cell death by targeting lipoylated TCA cycle proteins. *Science* **375**, 1254-1261 (2022).
24. Yoval-Sanchez B, Calleja LF, de la Luz Hernandez-Esquivel M, Rodriguez-Zavala JS. Piperlonguminine a new mitochondrial aldehyde dehydrogenase activator protects the heart from ischemia/reperfusion injury. *Biochim Biophys Acta Gen Subj* **1864**, 129684 (2020).
25. Jensen MT, *et al.* Heart rate at admission is a predictor of in-hospital mortality in patients with acute coronary syndromes: Results from 58 European hospitals: The European Hospital Benchmarking by Outcomes in acute coronary syndrome Processes study. *Eur Heart J Acute Cardiovasc Care* **7**, 149-157 (2018).
26. Delgado-Betancourt V, *et al.* Heart rate reduction after genetic ablation of L-type Ca(v)1.3 channels induces cardioprotection against ischemia-reperfusion injury. *Front Cardiovasc Med* **10**, 1134503 (2023).
27. Heusch G. Heart rate in the pathophysiology of coronary blood flow and myocardial ischaemia: benefit from selective bradycardic agents. *Br J Pharmacol* **153**, 1589-1601 (2008).
28. Scarsoglio S, Gallo C, Saglietto A, Ridolfi L, Anselmino M. Impaired coronary blood flow at higher heart rates during atrial fibrillation: Investigation via multiscale modelling. *Comput Methods Programs Biomed* **175**, 95-102 (2019).
29. Alzoubi KH, Al Mosabih HS, Mahasneh AF. The protective effect of edaravone on memory impairment induced by chronic sleep deprivation. *Psychiatry Res* **281**, 112577 (2019).
30. Alzoubi KH, Mayyas F, Abu Zamzam HI. Omega-3 fatty acids protects against chronic sleep-deprivation induced memory impairment. *Life Sci* **227**, 1-7 (2019).
31. Powell SR, Hall D, Shih A. Copper loading of hearts increases postischemic reperfusion injury. *Circ Res* **69**, 881-885 (1991).
32. Bar-Or D, McDonald MC, Thiernemann C. Reduction of infarct size in a rat model of regional myocardial ischemia and reperfusion by the synthetic peptide DAHK. *Crit Care Med* **34**, 1955-1959 (2006).
33. Khaliulin I, Schneider A, Houminer E, Borman JB, Schwalb H. Apomorphine prevents myocardial ischemia/reperfusion-induced oxidative stress in the rat heart. *Free Radic Biol Med* **37**, 969-976 (2004).
34. Delorey PE. Coping with change. *NLN Publ*, 83-91 (1976).

35. Tan W, *et al.* Hsp90 Inhibitor STA9090 induced VPS35 related extracellular vesicle release and metastasis in hepatocellular carcinoma. *Transl Oncol* **26**, 101502 (2022).
36. Maiti S, Picard D. Cytosolic Hsp90 Isoform-Specific Functions and Clinical Significance. *Biomolecules* **12**, (2022).
37. Lu D, *et al.* FOXO3a-dependent up-regulation of HSP90 alleviates cisplatin-induced apoptosis by activating FUNDC1-mediated mitophagy in hypoxic osteosarcoma cells. *Cell Signal* **101**, 110500 (2023).
38. Li Z, *et al.* Targeting mitochondria-inflammation circle by renal denervation reduces atheroprone endothelial phenotypes and atherosclerosis. *Redox Biol* **47**, 102156 (2021).
39. Sharp TE, 3rd, *et al.* Renal Denervation Prevents Heart Failure Progression Via Inhibition of the Renin-Angiotensin System. *J Am Coll Cardiol* **72**, 2609-2621 (2018).

REVIEWERS' COMMENTS

Reviewer #1 (Remarks to the Author):

Thank you for your revision. The manuscript is improved.

Summary Statement: Chen and colleagues examined the impact sleep fragmentation on myocardial injury in the male rat model and examined the role of sympathetic nerve activity mediated cardiomyocyte copper overload as a mechanism in cardiomyocyte apoptosis. The authors have greatly improved the present manuscript based upon reviewer comments. However, one minor methodological concern remains in reference HRV parameters and associated conclusions. Please see the minor comments below to improve/finalize the present manuscript.

MINOR COMMENTS

Methods

1. The methodology and conclusions associated with HRV is improved, but still not quite accurate. Normalized HF is calculated by dividing HF by total power of HF and LF, so inherently, the unreliable LF band is still included. I would instead focus on raw HF power. Further, a reduction in HF is associated with a reduction in parasympathetic innervation of the heart. Please do not comment on sympathetic drive associated with this parameter. I know this is bit nitpicky. However, your whole story of presumed reduced HF HRV and an increase in cardiac NE suggests a shift in autonomic balance.

Reviewer #2 (Remarks to the Author):

The authors have adequately addressed my initial comments by adding new experiments to support and strengthen their conclusion and by rewriting part of the manuscript (adding also new references).

Reviewer #3 (Remarks to the Author):

The authors have comprehensively addressed all the concerns that arose in the first review. They performed additional clarifying experiments and gave comprehensive replies also regarding comments raised by results obtained by other authors. Even from a formal/theoretical point of view, the figures are now accurate and well represent a mechanism of damage to the cardiac tissue mediated by the accumulation of copper. In my opinion, these results deserve publication, are of great relevance, and could stimulate future clinical studies

Reviewer #1

Summary Statement: Chen and colleagues examined the impact sleep fragmentation on myocardial injury in the male rat model and examined the role of sympathetic nerve activity mediated cardiomyocyte copper overload as a mechanism in cardiomyocyte apoptosis. The authors have greatly improved the present manuscript based upon reviewer comments. However, one minor methodological concern remains in reference HRV parameters and associated conclusions. Please see the minor comments below to improve/finalize the present manuscript.

MINOR COMMENTS

Methods

1. The methodology and conclusions associated with HRV is improved, but still not quite accurate. Normalized HF is calculated by dividing HF by total power of HF and LF, so inherently, the unreliable LF band is still included. I would instead focus on raw HF power. Further, a reduction in HF is associated with a reduction in parasympathetic innervation of the heart. Please do not comment on sympathetic drive associated with this parameter. I know this is bit nitpicky. However, your whole story of presumed reduced HF HRV and an increase in cardiac NE suggests a shift in autonomic balance.

Our response:

We deeply apologize for any previous misunderstanding of your intentions, and we once again express our gratitude for your meticulous and patient clarification. We have replaced HFnorm with HF and accordingly revised the description of the results (Fig.3J, line 114 to 116, page 6). We sincerely appreciate your valuable feedback and the time you invested in reviewing our manuscript. Your constructive comments have been crucial in guiding our revisions and improving the overall quality of our work.

Reviewer #2 (Remarks to the Author):

The authors have adequately addressed my initial comments by adding new experiments to support and strengthen their conclusion and by rewriting part of the manuscript (adding also new references).

Our response:

We are deeply grateful for your thoughtful and constructive feedback on our manuscript. Your

suggestions have significantly improved the clarity and depth of our work. We are pleased to hear that you have accepted our revisions and agree with the publication of our paper. Your support and expert guidance through the review process are highly appreciated.

Reviewer #3 (Remarks to the Author):

The authors have comprehensively addressed all the concerns that arose in the first review. They performed additional clarifying experiments and gave comprehensive replies also regarding comments raised by results obtained by other authors. Even from a formal/theoretical point of view, the figures are now accurate and well represent a mechanism of damage to the cardiac tissue mediated by the accumulation of copper. In my opinion, these results deserve publication, are of great relevance, and could stimulate future clinical studies.

Our response:

Thank you very much for your insightful comments. We are delighted to learn that you have accepted our responses and agree that our work is now ready for publication. Your detailed feedback was instrumental in refining our study and we are grateful for your expertise throughout the review process.